# A Tight Theory of Error Feedback Algorithms in Distributed Optimization

**Daniel Berg Thomsen** [1][2]  **Adrien Taylor** [1]  **Aymeric Dieuleveut** [2]

## Abstract

Communication costs are a major bottleneck in distributed learning and first-order optimization. A common approach to alleviate this issue is to compress the gradient information exchanged between agents. However, such compression typically degrades the convergence guarantees of gradient-based methods. *Error feedback* mechanisms provide a simple and computationally cheap remedy for this issue, but numerous variants have been proposed, and their relative performance remains poorly understood. This paper provides *tight* convergence analyses for two of the main error-feedback algorithms from the literature, the classic Error Feedback method (EF) and Error Feedback 21 (EF[21]), by identifying optimal step-size choices and constructing optimal Lyapunov functions tailored to each method. The results hold independently of the number of agents and recover the known best guarantees possible in the single-agent regime.

## 1. Introduction

The trend toward larger model usage in machine learning has made training in distributed environments a practical necessity. In many applications, including federated learning, data are partitioned across $n$ agents and a central server coordinates the process (McMahan et al., 2017; Kairouz et al., 2019). Consider the finite-sum problem

$$\min_{x \in \mathbb{R}^d} \left[ f(x) := \frac{1}{n} \sum_{i=1}^{n} f^{(i)}(x) \right], \qquad (1)$$

where only agent $i$ has access to first-order information about $f^{(i)}$. Communication follows a star topology: at each round, agents send messages to the server, the server aggregates them into a new iterate and broadcasts this iterate

[1]Inria, D.I. ENS, CNRS, PSL Research University, Paris, France [2]CMAP, École Polytechnique, Institut Polytechnique de Paris, Palaiseau, France. Correspondence to: Daniel Berg Thomsen <daniel.berg-thomsen@inria.fr>.

*Proceedings of the 43ʳᵈ International Conference on Machine Learning*, Seoul, South Korea. PMLR 306, 2026. Copyright 2026 by the author(s).

to all agents. Because many agents communicate concurrently, the agents-to-server traffic is often the dominant bottleneck (Seide et al., 2014; Chilimbi et al., 2014; Strom, 2015), especially for large models seen in deep learning.

A standard approach to mitigate communication costs is to reduce the frequency of communication (McMahan et al., 2017; Karimireddy et al., 2020; Mishchenko et al., 2022) and/or to transmit *compressed* messages. Compression may be applied on the uplink (agents→server) (Seide et al., 2014; Alistarh et al., 2017; Richtárik et al., 2021) and/or downlink (server→agents) (Harrane et al., 2018; Philippenko & Dieuleveut, 2020; Gorbunov et al., 2020); the focus here is on uplink compression. Common compressors include low-precision quantization (Alistarh et al., 2017), sparsification (e.g., Top-$K$, Alistarh et al. 2018), and using random projections (Vempala, 2004) like in SKETCHED-SGD (Ivkin et al., 2019).

To analyze algorithms independently of a particular compressor, one typically assumes general properties of a (possibly random) compression operator $\mathcal{C}$. Compression is modeled as a family of deterministic maps indexed by a seed $\omega$, written $\mathcal{C}(\cdot; \omega) : \mathbb{R}^d \to \mathbb{R}^d$. Unless stated otherwise, new seeds are drawn for the compression calls at each iteration; at a given iteration, the same seed may be used by several agents. When needed, $\omega_k^{(i)}$ denotes the seed used by agent $i$ in the compression step at iteration $k$, and $\omega_k := (\omega_k^{(i)})_{i=1}^{n}$; expectations are taken with respect to the relevant seeds. A classical assumption is unbiasedness, i.e., $\mathbb{E}_\omega[\mathcal{C}(x; \omega)] = x$ for all $x \in \mathbb{R}^d$. Another widely used assumption is *contractiveness*:

**Assumption 1.1** (Contractive compressor)**.** The compression operator $\mathcal{C}(\cdot; \omega)$ is such that, for some $\epsilon \in [0, 1)$,

$$\text{for all} \quad x \in \mathbb{R}^d, \quad \mathbb{E}_\omega \left[ \|x - \mathcal{C}(x; \omega)\|^2 \right] \le \epsilon \|x\|^2.$$

A natural baseline is *Compressed Gradient Descent* (CGD):

$$x_{k+1} := x_k - \frac{\eta}{n} \sum_{i=1}^{n} \mathcal{C}(\nabla f^{(i)}(x_k); \omega_k^{(i)}),$$

where $\eta > 0$ is the step size. However, CGD generally fails to converge in the multi-agent setting, as shown for instance when $\mathcal{C}$ is biased in Beznosikov et al. (2023).

To mitigate the effects of compression, each agent can use *Error Feedback* (EF). The classic mechanism (Seide et al., 2014; Karimireddy et al., 2019) keeps track of past compression errors and "reinjects" them into later messages, as described in Algorithm 1.

Another challenge in distributed environments is heterogeneity across agents: the local objectives $f^{(i)}$ can differ substantially, for example due to mismatched data distributions or uneven scaling induced by non-uniform data partitioning. Such heterogeneity can obstruct convergence. In the strongly convex setting, Beznosikov et al. (2023) prove linear convergence of distributed EF with biased compression, but only under the additional assumption that all local objectives share the same minimizer—i.e., the interpolation regime (Ma et al., 2018; Vaswani et al., 2019).

Motivated by the need to directly handle heterogeneity, Richtárik et al. (2021) proposed *Error Feedback 21* (EF$^{21}$), given in Algorithm 2. In EF$^{21}$, each agent maintains an estimator $d_k^{(i)}$ of the local gradient. At iteration $k$, the server first updates $x_{k+1}$ using the current estimators $d_k^{(i)}$. The agents then communicate compressed differences $\mathcal{C}(\nabla f^{(i)}(x_{k+1}) - d_k^{(i)}; \omega_k^{(i)})$, which are used to update the estimators for the next iteration. This tracking step is argued to be more stable around the global optimum of the sum total objective (1).

By now, a substantial literature has been devoted to the analysis of both algorithms under different assumptions on the communication model (Koloskova et al., 2019; Philippenko & Dieuleveut, 2021), the compression operator (Alistarh et al., 2017; Stich et al., 2018; Beznosikov et al., 2023), and the objective functions (Karimireddy et al., 2019; Stich & Karimireddy, 2020; Richtárik et al., 2021). The vast literature on the subject is further complicated by the many variants of these methods that have been proposed, see e.g., Zheng et al. (2019); Li & Li (2022); Tang et al. (2021); Fatkhullin et al. (2023); Tian et al. (2026); Condat et al. (2022); Fatkhullin et al. (2025); Gruntkowska et al. (2025); Egger et al. (2025); Redie et al. (2026), among others.

Many of the above papers provide *theoretical guarantees*, most often in the form of convergence rates that upper-bound the value of a hand-crafted metric. Yet these bounds can be loose, the analysis pessimistic, or the metric itself poorly matched to the phenomenon of interest, which complicates meaningful comparisons between algorithms. Building on recent advances in computer-aided proofs for optimization and automated construction of Lyapunov functions, Berg Thomsen et al. (2025) derived *tight* convergence rates for CGD, EF, and EF$^{21}$. Their rates are optimistic compared to existing results and give a *definitive* characterization of the worst-case convergence of these methods on smooth and strongly convex functions; notably, they also show that EF and EF$^{21}$ attain the same optimized worst-case rate. However, this analysis is restricted to the single-agent setting $n = 1$, which remains a significant theoretical and practical limitation.

## 1.1. Contributions.

In this paper, we provide *tight analyses* of both EF and EF$^{21}$ in the multi-agent setting, $n > 1$, under contractive compression (Assumption 1.1) and strong convexity and smoothness of the individual objectives. The formal guarantees cover the regimes stated in the detailed contributions below. Outside those regimes, our general-heterogeneity claims are stated as empirically validated laws. We show that the behavior in the distributed setting can differ from the single agent case, in particular due to *heterogeneity* between agents. We distinguish between two types of heterogeneity: statistical heterogeneity (local minimizers can differ across agents) and heterogeneous regularity parameters (local smoothness and strong-convexity constants can differ across agents).

These rates are *tight* in the method-specific sense used throughout the paper: within the stated state and candidate Lyapunov classes, they identify the *best tuning*, the *best Lyapunov certificate*, and the *smallest worst-case contraction factor*. They should not be read as method-agnostic communication-complexity lower bounds over all compressed distributed methods. To obtain them, we combine advanced proof techniques with numerical evaluations that corroborate the theoretical predictions. Altogether, our results provide a systematic picture of the worst-case behavior of these methods in the regimes covered by our formal guarantees, complemented by empirical laws for the general heterogeneous case.

**Rigor: theorems, certificates, and empirical laws.** Most of our results are stated either as a theorem with a complete proof or as a counterexample. To facilitate verification, we also attach **proof certificates** to each formal statement, which serve as either analytical or numerical validation of the stated guarantee. Specifically, certificates of correctness are provided either

(i) by a *Computer Algebra System* (CAS) via a Wolfram Language script, or

(ii) by numerically solving the associated *Performance Estimation Problems* (PEP).

CAS certificates verify algebraic identities, whereas PEP certificates provide numerical confirmation of complete statements. In the paper, the CAS and PEP certificate badges indicate the available certificates and link directly to the corresponding Wolfram Language script or Jupyter notebook

in the public GitHub repository.[1]

For cases in which a formal proof could not be identified, we state certain results as **empirical laws** instead. These laws are supported by extensive numerical evidence indicating that a proposed closed-form expression matches, up to numerical precision, the corresponding analytical quantity, which may itself lack a closed-form representation.

**Detailed contributions and outline.** More precisely, we make the following contributions, summarized in Table 1.

(i) In Section 2, we exhibit simple quadratic counterexamples showing that, for contractive compressors, both CGD and classic EF exhibit cycles in heterogeneous multi-agent settings and therefore do not converge in general.

(ii) For $\text{EF}^{21}$ under contractive compression and statistical heterogeneity, we derive in Subsection 3.1 a tight Lyapunov function, optimal step size, and worst-case contraction factor, and the optimal rate is shown to be independent of the number of agents.

(iii) For deterministic, additive, positively homogeneous compressors with heterogeneous regularity parameters, we also prove a sharp linear convergence guarantee for both $\text{EF}^{21}$ and EF with averaged parameters $(\bar{L}, \bar{\mu})$.

(iv) In Subsection 3.2, we construct a tight Lyapunov function for classic EF under statistical homogeneity, and show that its optimal step size and rate coincide with those of $\text{EF}^{21}$, recovering the single-agent results for any $n$.

(v) Finally, in Section 4, we state *empirical laws* that characterize the optimal step size for EF and $\text{EF}^{21}$ under general heterogeneity, an optimal Lyapunov function for $\text{EF}^{21}$, an explicit polynomial rate formula for the $n = 2$ case, and an optimal two-parameter tuning rule for EControl (Gao et al., 2023), supported by extensive numerical verification via numerical solution of the corresponding performance estimation problems.

In the next section, we introduce the algorithms, assumptions and counterexamples.

## 2. Background

This section motivates our multi-agent analysis by showing that classic EF can fail under heterogeneity unless additional structure is imposed. Specifically, Subsection 2.1

provides counterexamples, and Subsection 2.2 reviews existing theoretical results on EF and $\text{EF}^{21}$. The definitions and notation needed to state the main results of this paper are provided in Subsections 2.3 and 2.4.

### 2.1. Non-Convergence under Statistical Heterogeneity

The following counterexamples demonstrate that neither CGD nor EF can achieve arbitrary accuracy in the presence of statistical heterogeneity.

**Compressed Gradient Descent.** Consider the case where there are $n = 2$ agents, each having access to first-order oracles querying the one-dimensional quadratic functions

$$f^{(1)}(x) := \frac{\mu}{2}x^2 - x, \qquad f^{(2)}(x) := \frac{\mu}{2}x^2 + x, \quad (2)$$

where $\mu > 0$ is a constant. These functions are $\mu$-strongly convex and $L$-smooth for any $L > \mu$. Set

$$x_0 := \eta \frac{\sqrt{\epsilon}}{2 - \eta\mu}. \quad (3)$$

By definition, for CGD

$$x_1 = x_0 - \frac{\eta}{2}\left[\mathcal{C}((f^{(1)})'(x_0); \omega_0^{(1)}) + \mathcal{C}((f^{(2)})'(x_0); \omega_0^{(2)})\right]. \quad (4)$$

Under Assumption 1.1, the compression oracle may respond

$$\mathcal{C}((f^{(1)})'(x_0); \omega_0^{(1)}) = (1 - \sqrt{\epsilon})(f^{(1)})'(x_0),$$
$$\mathcal{C}((f^{(2)})'(x_0); \omega_0^{(2)}) = (1 + \sqrt{\epsilon})(f^{(2)})'(x_0).$$

Plugging this, the derivatives of $f^{(1)}$ and $f^{(2)}$, and the definition of $x_0$ into (4),

$$x_1 = x_0 - \frac{\eta}{2}\left[2\mu x_0 + 2\sqrt{\epsilon}\right] = x_0(1 - \eta\mu) - \eta\sqrt{\epsilon}$$
$$= \frac{\eta\sqrt{\epsilon}}{2 - \eta\mu}\left[(1 - \eta\mu) - (2 - \eta\mu)\right] = -x_0.$$

By symmetry, if the compression oracle next responds with

$$\mathcal{C}((f^{(1)})'(x_1); \omega_1^{(1)}) = (1 + \sqrt{\epsilon})(f^{(1)})'(x_1),$$
$$\mathcal{C}((f^{(2)})'(x_1); \omega_1^{(2)}) = (1 - \sqrt{\epsilon})(f^{(2)})'(x_1).$$

then the same computation gives $x_2 = x_0$, yielding a 2-step cycle.

**Error Feedback.** Consider now exactly the same functions defined in (2). The following proposition shows that the same behavior can be observed in EF.

**Proposition 2.1.** *Let there be $n = 2$ agents, each having access to first-order oracles querying the one-dimensional quadratic functions defined in (2). Then, for each fixed step size $\eta > 0$ in the following cases, there exist admissible compressor responses under Assumption 1.1 for which the full EF state $(x_k, e_k^{(1)}, e_k^{(2)})$ is 2-periodic:*

---

[1]These certificates do not replace the mathematical proofs in the paper; rather, they provide an additional layer of transparency and error checking, analogous to unit tests in software development. They offer a reproducible, independently verifiable record supporting the theoretical claims and help reduce the risk of oversights in complex derivations.

| Agents | Heterogeneity | EF | EF²¹ | EControl |
|---|---|---|---|---|
| $n = 1$ | — | Berg Thomsen et al. (2025) | | |
| $n > 1$ | None | Theorem 3.4 `PEP` `CAS` | Theorem 3.1 `PEP` `CAS` | Empirical Law 4.4 `PEP` |
| | Statistical† | *Counterexample* 2-cycle (Proposition 2.1) | | |
| | Regularity parameters‡ | Corollary 3.2 (Linear)* | | |
| $n = 2$ | Both†‡ | Empirical Law 4.3 `PEP` | | |

*Table 1.* Summary of convergence results and empirical tuning laws for EF²¹, EF, and EControl. † Statistical heterogeneity means that the local minimizers $x_\star^{(i)}$ are not identical; ‡ heterogeneous regularity parameters mean that the local smoothness and strong-convexity constants are not identical; ∗ the linear corollary additionally assumes (beyond contractive compression) that $\mathcal{C}$ is deterministic, additive, and positively homogeneous ($\mathcal{C}(x + y) = \mathcal{C}(x) + \mathcal{C}(y)$, $\mathcal{C}(\alpha x) = \alpha \mathcal{C}(x)$ for $\alpha \geq 0$). Heterogeneous guarantees for EF²¹ and EF are currently limited to linear compressors and the $n = 2$ empirical law. PEP and CAS badges link to certificates where available.

---

**Algorithm 1** Classic error feedback — EF

1: **initialization:** $x_0 \in \mathbb{R}^d, \eta > 0, e_0^{(i)} = 0$ for each $i \in [n]$; seeds $(\omega_k^{(i)})_{k \geq 0}$ given
2: **for** $k = 0, 1, 2, \ldots, K - 1$ **do**
3:     Each agent $i \in [n]$ compresses $e_k^{(i)} + \eta \nabla f^{(i)}(x_k)$ and communicates $m_k^{(i)} := \mathcal{C}(e_k^{(i)} + \eta \nabla f^{(i)}(x_k); \omega_k^{(i)})$
4:     Each agent $i \in [n]$ updates $e_{k+1}^{(i)} \leftarrow e_k^{(i)} + \eta \nabla f^{(i)}(x_k) - m_k^{(i)}$
5:     Server updates $x_{k+1} \leftarrow x_k - \frac{1}{n} \sum_{i=1}^n m_k^{(i)}$
6: **end for**

---

**Algorithm 2** Error Feedback 21 — EF²¹

1: **initialization:** $x_0 \in \mathbb{R}^d$; step size $\eta > 0$; seeds $(\omega_k^{(i)})_{k \geq -1}$ given; $d_0^{(i)} = \mathcal{C}(\nabla f^{(i)}(x_0); \omega_{-1}^{(i)})$ for each $i \in [n]$;
2: **for** $k = 0, 1, 2, \ldots, K - 1$ **do**
3:     Server updates $x_{k+1} \leftarrow x_k - \eta \cdot \frac{1}{n} \sum_{i=1}^n d_k^{(i)}$
4:     Each agent $i \in [n]$ compresses $\nabla f^{(i)}(x_{k+1}) - d_k^{(i)}$ and communicates $m_k^{(i)} := \mathcal{C}(\nabla f^{(i)}(x_{k+1}) - d_k^{(i)}; \omega_k^{(i)})$
5:     Each agent $i \in [n]$ updates $d_{k+1}^{(i)} \leftarrow d_k^{(i)} + m_k^{(i)}$
6: **end for**

---

1. $\eta < \frac{2}{\mu}$, with $x_0 = \eta \frac{\sqrt{\epsilon}}{2 - \eta \mu}$.

2. $\eta > \frac{2}{\mu}$, with $x_0 = -\eta \frac{\sqrt{\epsilon}}{2 - \eta \mu}$.

3. $\eta = \frac{2}{\mu}$, with any $x_0$.

A simple proof of this is provided in Appendix A.1.

**2.2. Related Work**

Error feedback has a long history in signal processing, where it is used to compensate for quantization in communication (Cutler, 1952; Inose & Yasuda, 2005). In distributed optimization, it dates back to the classical work of Seide et al. (2014). Following its introduction, subsequent work has studied the behavior of error feedback under various assumptions, including analyses for specific compression operators such as deterministic Top-$K$ (Alistarh et al., 2018) and stochastic compressors (Wu et al., 2018).

A complementary line of work leverages the contractive properties of compression operators to derive general convergence guarantees for a broad class of operators. In this setting, it is common to distinguish between *biased* and *unbiased* compressors (Beznosikov et al., 2023). Convergence

rates for error feedback with contractive compressors have been established for strongly convex functions (Stich et al., 2018), quasi-convex and nonconvex functions (Karimireddy et al., 2019), and using stochastic gradients (Stich & Karimireddy, 2020). Tight rates have also been established under the same assumptions as this work, for the single-agent regime (Berg Thomsen et al., 2025).

Error Feedback 21 (EF²¹) (Richtárik et al., 2021) is a variant of error feedback that was designed specifically to handle the heterogeneity of the multi-agent setting. The same work also reports experiments on logistic regression with a nonconvex regularizer. EF²¹ and its variants have been studied in many different scenarios, some of which include using stochastic gradients, momentum (Fatkhullin et al., 2023) and practical extensions such as bidirectional compression, variance reduction, and proximal setups (Fatkhullin et al., 2021).

A related two-parameter method is EControl (Gao et al., 2023), which introduces a controllable error-compensation mechanism combining ideas from EF and EF²¹. It is shown to converge in strongly convex, convex, and nonconvex settings.

## 2.3. Assumptions and Notations

The following definition is used throughout this work.

**Definition 2.2** (Class $\mathcal{F}_{\mu,L}$)**.** For constants $0 < \mu < L$, denote by $\mathcal{F}_{\mu,L}$ the set of functions $h : \mathbb{R}^d \to \mathbb{R}$ that are $L$-smooth and $\mu$-strongly convex. That is, for any $h \in \mathcal{F}_{\mu,L}$, and any $x, y \in \mathbb{R}^d$, it holds that

$$h(y) \le h(x) + \langle \nabla h(x), y - x \rangle + \frac{L}{2} \|y - x\|^2,$$

and

$$h(y) \ge h(x) + \langle \nabla h(x), y - x \rangle + \frac{\mu}{2} \|y - x\|^2.$$

Throughout this work, each local function $f^{(i)}$ is assumed to belong to $\mathcal{F}_{\mu^{(i)}, L^{(i)}}$.

The symbol $\mathbb{S}^\ell$ denotes the set of symmetric matrices, and $\mathbb{S}^\ell_+$ denotes the set of positive semidefinite matrices. For any two matrices $A \in \mathbb{S}^\ell$ and $B \in \mathbb{S}^d$, the Kronecker product is denoted by $A \otimes B$. The condition number is denoted by $\kappa := \frac{L}{\mu}$. For any objective function $f \in \mathcal{F}_{\mu,L}$, the minimizer is denoted by $x_\star := \arg\min_{x \in \mathbb{R}^d} f$, and its minimum value is denoted by $f_\star := \min_{x \in \mathbb{R}^d} f(x)$. For each local function $f^{(i)}$, its unique minimizer is denoted by $x_\star^{(i)}$ and its minimum value by $f_\star^{(i)}$.

## 2.4. Methodology

The analysis contained in Section 3 relies on the systematic identification of Lyapunov functions that provide a tight convergence rate for each method. These Lyapunov functions are *optimal* with respect to a large class of Lyapunov functions, defined in this section.

**Lyapunov functions.** Let $\mathcal{M} : (\mathbb{R}^d)^\ell \times \mathbb{R}^d \times \mathcal{F} \to (\mathbb{R}^d)^\ell \times \mathbb{R}^d$ denote a first-order method acting on a set of functions $\mathcal{F}$ of dimension $d$, for an integer $\ell \in \mathbb{N}$ different *state variables*. Such a method, given a function $f \in \mathcal{F}$, is applied to an initial *state* $\xi_0 \in (\mathbb{R}^d)^\ell$ and iterate $x_0 \in \mathbb{R}^d$, and generates the next state $\xi_1$ and next iterate $x_1$. The *states* represent information summarizing the current point in the optimization trajectory, including auxiliary quantities that the algorithms may depend on—for example, error-related quantities in error feedback algorithms.

**Definition 2.3** (Candidate Lyapunov function)**.** A function $\mathcal{V} : (\mathbb{R}^d)^\ell \times \mathbb{R}^d \times \mathcal{F} \to \mathbb{R}$, written $\mathcal{V}(\xi, x; f)$, is called a candidate Lyapunov function if, for every $f \in \mathcal{F}$, it satisfies the following conditions:

1. (Non-negativity) $\mathcal{V}(\xi, x; f) \ge 0$, for any $\xi \in (\mathbb{R}^d)^\ell$, $x \in \mathbb{R}^d$,

2. (Zero at fixed-point) $\mathcal{V}(\xi, x; f) = 0$ if and only if $x = x_\star$ and $\xi = \xi_\star$ for a unique $\xi_\star \in (\mathbb{R}^d)^\ell$.

3. (Meaningful lower bound) there exists a positive semidefinite matrix $P \in \mathbb{S}^\ell_+$ and a scalar $p \ge 0$ such that, for all $\xi \in (\mathbb{R}^d)^\ell$ and $x \in \mathbb{R}^d$,

$$\mathcal{V}(\xi, x; f) \ge (\xi - \xi_\star)^\top (P \otimes I_d)(\xi - \xi_\star) + p(f(x) - f_\star),$$

and $\mathrm{Tr}(P) + p = 1$.

In quadratic forms involving $P \otimes I_d$, we identify $\xi = (\xi^{(1)}, \ldots, \xi^{(\ell)}) \in (\mathbb{R}^d)^\ell$ with the stacked vector $[(\xi^{(1)})^\top, \ldots, (\xi^{(\ell)})^\top]^\top \in \mathbb{R}^{\ell d}$, where each block $\xi^{(j)} \in \mathbb{R}^d$ is one state variable.

The objective is to find candidate Lyapunov functions $\mathcal{V}$ satisfying the recurrence

$$\mathcal{V}(\xi_1, x_1; f) \le \rho \cdot \mathcal{V}(\xi_0, x_0; f),$$

for some constant $\rho < 1$, uniformly over $\mathcal{F}$. Finding the *optimal* Lyapunov function within a parameterized class for a method $\mathcal{M}$ amounts to solving the following problem:

$$\rho_\star(\mathcal{M}) := \min_{\mathcal{V}} \quad \sup_{\substack{f \in \mathcal{F}_{\mu,L} \\ \xi_0, x_0 \\ \mathcal{V}(\xi_0, x_0; f) > 0}} \frac{\mathcal{V}(\xi_1, x_1; f)}{\mathcal{V}(\xi_0, x_0; f)} \tag{5}$$
$$\text{s.t. } (\xi_1, x_1) = \mathcal{M}(\xi_0, x_0; f).$$

The goal of this work is to identify optimal Lyapunov functions for EF and EF[21] and formally prove that they achieve the convergence rate defined in (5). It has been shown that optimal candidate Lyapunov functions can be identified by solving semidefinite programs (SDPs), yielding numerical convergence guarantees (Taylor et al., 2018). Control-inspired SDP analyses have also been used for decentralized optimization over communication graphs (Sundararajan et al., 2019; 2020).

We used numerical tools both to obtain guarantees (Taylor et al., 2017a; Goujaud et al., 2024) and to search for Lyapunov functions (Taylor et al., 2018; Upadhyaya et al., 2025). Such numerical evidence is not, on its own, a theoretical proof, but it can reveal stable certificate structures that can then be simplified and proved analytically. In the multi-agent setting, the optimal Lyapunov coefficients need not be unique, so Appendix B describes the SDP-guided Lyapunov-search heuristics used to infer closed-form structures.

---

**Algorithm 3** Error Control — EControl

1: **initialization:** $x_0 \in \mathbb{R}^d$; parameters $\eta \geq 0$ and $\gamma > 0$; $e_0^{(i)} = 0, d_0^{(i)} = 0$ for $i \in [n]$; seeds $(\omega_k^{(i)})_{k \geq 0}$ given
2: **for** $k = 0, 1, 2, \ldots, K - 1$ **do**
3:     Each agent $i \in [n]$ communicates the compressed message $m_k^{(i)} := \mathcal{C}(\eta e_k^{(i)} + \nabla f^{(i)}(x_k) - d_k^{(i)}; \omega_k^{(i)})$
4:     Each agent $i \in [n]$ updates $d_{k+1}^{(i)} \leftarrow d_k^{(i)} + m_k^{(i)}$
5:     Server updates $x_{k+1} \leftarrow x_k - \gamma \cdot \frac{1}{n} \sum_{i=1}^{n} d_{k+1}^{(i)}$
6:     Each agent $i \in [n]$ updates $e_{k+1}^{(i)} \leftarrow e_k^{(i)} + \nabla f^{(i)}(x_k) - d_{k+1}^{(i)}$
7: **end for**

---

## 3. Main Results

This section states tight convergence guarantees for $\text{EF}^{21}$ under contractive compression, a linear-compressor corollary with heterogeneous regularity parameters, and corresponding guarantees for EF under statistical homogeneity. In light of the counterexample in Proposition 2.1, the result for EF requires the additional assumption that the local objectives $f^{(i)}$ share the same minimizer.

### 3.1. Error Feedback 21

For $\text{EF}^{21}$, the state is defined as

$$\xi_k^{\text{EF}^{21}} := \begin{bmatrix} [x_k, \ldots, x_k]^\top \\ [\nabla f^{(1)}(x_k), \ldots, \nabla f^{(n)}(x_k)]^\top \\ [d_k^{(1)}, \ldots, d_k^{(n)}]^\top \end{bmatrix}. \quad (6)$$

Its fixed-point value is

$$\xi_\star^{\text{EF}^{21}} := \begin{bmatrix} [x_\star, \ldots, x_\star]^\top \\ [\nabla f^{(1)}(x_\star), \ldots, \nabla f^{(n)}(x_\star)]^\top \\ [\nabla f^{(1)}(x_\star), \ldots, \nabla f^{(n)}(x_\star)]^\top \end{bmatrix}.$$

Candidate Lyapunov functions may thus include many terms, but the Lyapunov function given in Theorem 3.1 is relatively simple, and is worst-case optimal under the optimal step size tuning.

**Theorem 3.1.** [CAS] [PEP]
*Let $\epsilon \in (0, 1)$ and assume that the compression operator $\mathcal{C}$ satisfies Assumption 1.1. Let $f^{(i)} \in \mathcal{F}_{\mu,L}$ for each $i \in [n]$. Let the step size be given by*

$$\eta^\star = \left( \frac{2}{L + \mu} \right) \cdot \left( \frac{1 - \sqrt{\epsilon}}{1 + \sqrt{\epsilon}} \right). \quad (7)$$

*Then, the Lyapunov function*

$$\mathcal{V}\left( \xi^{\text{EF}^{21}}, x; f \right) := \frac{\sqrt{\epsilon}}{n} \left\| \sum_{i=1}^{n} \nabla f^{(i)}(x_k) \right\|^2 \\ + \sum_{i=1}^{n} \| \nabla f^{(i)}(x_k) - d_k^{(i)} \|^2 \quad (8)$$

*is optimal (i.e., solves (5)) and satisfies*

$$\mathbb{E}_\omega \left[ \mathcal{V}\left( \xi_{k+1}^{\text{EF}^{21}}, x_{k+1}; f \right) \right] \leq \rho_\star \cdot \mathbb{E}_\omega \left[ \mathcal{V}\left( \xi_k^{\text{EF}^{21}}, x_k; f \right) \right]$$

*where the rate is given by*

$$\rho_\star := \sqrt{\epsilon} + \left( \frac{1 - \sqrt{\epsilon}}{2} \right) \left( \frac{\kappa - 1}{\kappa + 1} \right)^2 \Psi(\kappa, \epsilon) \quad (9)$$

*and*

$$\Psi(\kappa, \epsilon) := 1 - \sqrt{\epsilon} + \sqrt{(1 + \sqrt{\epsilon})^2 + \sqrt{\epsilon} \, 16 \frac{\kappa}{(\kappa - 1)^2}}.$$

*Finally, the step size in (7) is worst-case optimal for $\text{EF}^{21}$: within the candidate Lyapunov class based on $\xi^{\text{EF}^{21}}$, it achieves the minimal worst-case one-step contraction factor, $\rho_\star$ in (9). This bound is also multi-step tight, meaning that after $k$ iterations the worst-case contraction equals $\rho_\star^k$.*

A proof appears in Appendix A.2. The numerical and symbolic certificates can be accessed by clicking the PEP and CAS badges in the theorem header.

The symmetry across agents implies that the worst-case rate in Theorem 3.1 matches the single-agent rate: the worst-case instance for $n = 1$ can be replicated across agents. Though the single-agent Lyapunov function is recovered when setting $n = 1$, the multi-agent Lyapunov function is not a trivial extension of the single-agent case. One could have expected a weighted sum of the single-agent Lyapunov functions defined in (Berg Thomsen et al., 2025) to be sufficient, but it is clear that the weighting placed on the individual terms of (8) do not correspond to that.

The next corollary extends Theorem 3.1 to heterogeneous regularity parameters (agent-specific smoothness and strong-convexity constants) under deterministic, additive, positively homogeneous compression, a setup in which Richtárik et al. (2021) showed an equivalence between EF and $\text{EF}^{21}$, under a reparametrization.

**Corollary 3.2.**
*Let $\epsilon \in [0, 1)$ and assume that the compression operator $\mathcal{C}$ satisfies Assumption 1.1 and is deterministic, additive, and positively homogeneous (so $\mathcal{C}(x + y) = \mathcal{C}(x) + \mathcal{C}(y)$ and $\mathcal{C}(\alpha x) = \alpha \mathcal{C}(x)$ for all $\alpha \geq 0$). Let $f^{(i)} \in \mathcal{F}_{\mu^{(i)}, L^{(i)}}$ for each $i \in [n]$, and define*

$$\bar{L} := \frac{1}{n} \sum_{i=1}^{n} L^{(i)}, \qquad \bar{\mu} := \frac{1}{n} \sum_{i=1}^{n} \mu^{(i)}.$$

$$\kappa_\Sigma := \frac{\bar{L}}{\bar{\mu}} = \frac{\sum_{i=1}^{n} L^{(i)}}{\sum_{i=1}^{n} \mu^{(i)}}.$$

*Let the step size be given by*

$$\eta^\star = \left( \frac{2}{\bar{L} + \bar{\mu}} \right) \cdot \left( \frac{1 - \sqrt{\epsilon}}{1 + \sqrt{\epsilon}} \right).$$

*Then the Lyapunov function*

$$\mathcal{V}_{\text{lin}}\left(\xi^{\text{EF}^{21}}, x; f\right) := \sqrt{\epsilon}\, \|\bar{g}_k\|^2 + \|\bar{g}_k - \bar{d}_k\|^2$$

*satisfies the deterministic contraction*

$$\mathcal{V}_{\text{lin}}\left(\xi^{\text{EF}^{21}}_{k+1}, x_{k+1}; f\right) \leq \rho_\star \cdot \mathcal{V}_{\text{lin}}\left(\xi^{\text{EF}^{21}}_k, x_k; f\right),$$

*where $\bar{g}_k := \frac{1}{n}\sum_{i=1}^n \nabla f^{(i)}(x_k)$, $\bar{d}_k := \frac{1}{n}\sum_{i=1}^n d_k^{(i)}$, and the rate $\rho_\star$ is given by (9) with $\kappa = \kappa_\Sigma$.*

*Remark* 1. Under the assumptions of Corollary 3.2, distributed CGD also converges, since the method reduces to a single-agent compressed iteration on the averaged objective. The single-agent result of Berg Thomsen et al. (2025) then shows that CGD outperforms both EF and $\text{EF}^{21}$ in this regime.

A proof appears in Appendix A.3. The same step size and rate apply to EF, matching Theorems 1–2 in Berg Thomsen et al. (2025) evaluated at $\kappa_\Sigma$. The result follows by observing that the barred variables evolve exactly as a single-agent $\text{EF}^{21}$ instance with parameters $(\bar{L}, \bar{\mu})$, and then invoking the EF–$\text{EF}^{21}$ equivalence. Extending tight guarantees beyond this deterministic linear setting appears technically challenging, motivating the next section.

### 3.2. Classic Error Feedback under Statistical Homogeneity

Due to the counterexample given in Proposition 2.1, an additional assumption is required to prove convergence for EF. This is the same assumption as that required for the linear rate of convergence given in (Beznosikov et al., 2023)—namely, that the minimizers of the local functions are all the same, a condition commonly known as the interpolation regime (Ma et al., 2018; Vaswani et al., 2019).

**Assumption 3.3** (Statistical homogeneity). The local objectives satisfy *statistical homogeneity*, i.e., they share the same minimizer:

$$x_\star^{(i)} = x_\star^{(j)} \quad \text{for all } i, j \in [n].$$

For EF, the state is defined as

$$\xi_k^{\text{EF}} := \begin{bmatrix} x_k \\ [\nabla f^{(1)}(x_k), \ldots, \nabla f^{(n)}(x_k)]^\top \\ [m_k^{(1)}, \ldots, m_k^{(n)}]^\top \\ [e_k^{(1)}, \ldots, e_k^{(n)}]^\top \end{bmatrix}.$$

Its fixed-point value is

$$\xi_\star^{\text{EF}} := \begin{bmatrix} x_\star \\ [0, \ldots, 0]^\top \\ [0, \ldots, 0]^\top \\ [0, \ldots, 0]^\top \end{bmatrix},$$

where $\nabla f^{(i)}(x_\star) = 0$ for all $i$ under Assumption 3.3, so the communication and error-feedback blocks are also zero at the fixed point. Similar to Theorem 3.1, Theorem 3.4 provides a relatively simple Lyapunov function which is worst-case optimal under the optimal step size tuning.

**Theorem 3.4.** `CAS` `PEP`
*Let $\epsilon \in (0, 1)$ and assume that the compression operator $\mathcal{C}$ satisfies Assumption 1.1. Let $f^{(i)} \in \mathcal{F}_{\mu,L}$ for each $i \in [n]$, and assume Assumption 3.3.*

*Let the step size be given by $\eta^\star$ as defined in (7). Then, the Lyapunov function*

$$\mathcal{V}\left(\xi^{\text{EF}}, x; f\right) := \frac{1}{n\sqrt{\epsilon}} \left\| \sum_{i=1}^n e_k^{(i)} \right\|^2 + \sum_{i=1}^n \|x_k - x_\star - e_k^{(i)}\|^2. \tag{10}$$

*is optimal and satisfies*

$$\rho_\star(\text{EF}) = \rho_\star,$$

*where $\rho_\star$ is defined in (9). Finally, the step size in (7) is worst-case optimal for EF: within the candidate Lyapunov class based on $\xi^{\text{EF}}$, it achieves the minimal worst-case one-step contraction factor, $\rho_\star$ in (9). This bound is also multi-step tight, meaning that after $k$ iterations the worst-case contraction equals $\rho_\star^k$.*

A proof appears in Appendix A.4. Like in the case of $\text{EF}^{21}$, the single-agent results from (Berg Thomsen et al., 2025) are recovered for any $n \geq 1$, including the optimal step size and the Lyapunov function (by setting $n = 1$ in (10)).

Under shared regularity parameters $f^{(i)} \in \mathcal{F}_{\mu,L}$, the worst-case contraction is independent of the number of agents $n$ for $\text{EF}^{21}$ without requiring a shared minimizer, and for classic EF under the shared-minimizer condition of Assumption 3.3. In both cases, the class-optimal step sizes and contraction factors coincide with their single-agent counterparts. In these distributed regimes, the corresponding single-agent analyses therefore characterize the worst-case rates.

## 4. Empirical Laws with Heterogeneous Regularity Parameters

The results of the previous sections either assume homogeneous regularity parameters (shared parameters $(L, \mu)$) or rely on linear deterministic compressors to handle heterogeneous regularity parameters. When this is relaxed to allow general heterogeneous regularity parameters, with agent-specific parameters $f^{(i)} \in \mathcal{F}_{\mu^{(i)}, L^{(i)}}$, numerical evidence indicates that the worst-case behavior depends on the local smoothness and strong-convexity parameters. Based on extensive numerical experiments, this section states empirical laws for (i) the optimal step size for both EF and $\text{EF}^{21}$, (ii)

a corresponding Lyapunov function for $\text{EF}^{21}$, (iii) the optimal convergence rate when $n = 2$, and (iv) a two-parameter tuning rule for EControl (Gao et al., 2023) in the general case.

The empirical laws below are formulated based on numerical validation using the Performance Estimation Problem (PEP) framework (Drori, 2014; Taylor et al., 2017b). Appendix C gives the verification details, including exact grid searches and parameter sweeps. The empirical formulas were tested across a wide range of problem parameters.

**Empirical Law 4.1** (General optimal step size). [PEP]
*Consider the setting with heterogeneous regularity parameters, i.e., let $f^{(i)} \in \mathcal{F}_{\mu^{(i)}, L^{(i)}}$. The optimal step size for both* EF *and* $\text{EF}^{21}$ *is given by*

$$\eta^\star = \left( \frac{2n}{\sum_{i=1}^n (L^{(i)} + \mu^{(i)})} \right) \cdot \left( \frac{1 - \sqrt{\epsilon}}{1 + \sqrt{\epsilon}} \right).$$

This step size reduces to (7) when all agents share the same parameters.

Numerical evidence further suggests that the following Lyapunov function is optimal for $\text{EF}^{21}$ among Lyapunov functions built from the class defined by (6).

**Empirical Law 4.2** (General Lyapunov function). [PEP]
*Let $S := \sum_{i=1}^n (L^{(i)} + \mu^{(i)})$ and $w_i := \frac{S}{L^{(i)} + \mu^{(i)}}$. Let each agent have heterogeneous regularity parameters, i.e., $f^{(i)} \in \mathcal{F}_{\mu^{(i)}, L^{(i)}}$. The optimal Lyapunov function for $\text{EF}^{21}$ is given by*

$$\mathcal{V}\left( \xi^{\text{EF}^{21}}, x; f \right) := \frac{1}{n} \sum_{i=1}^n w_i \| \nabla f^{(i)}(x_k) - d_k^{(i)} \|^2$$
$$+ \frac{\sqrt{\epsilon}}{n} \left\| \sum_{i=1}^n \nabla f^{(i)}(x_k) \right\|^2$$

This family of Lyapunov functions reduces to (8) when all agents share the same parameters; in the heterogeneous case, it weights each local estimator error by $\sum_{j=1}^n (L^{(j)} + \mu^{(j)})/(L^{(i)} + \mu^{(i)})$.

Finally, numerical evidence suggests an explicit characterization of the agent-sensitive convergence rate for both EF and $\text{EF}^{21}$ when $n = 2$:

**Empirical Law 4.3** (General rate for $n = 2$). [PEP]
*Let $n = 2$ and let the agents have heterogeneous regularity parameters, i.e., $f^{(i)} \in \mathcal{F}_{\mu^{(i)}, L^{(i)}}$. At the step size $\eta = \eta^\star$ from Empirical Law 4.1, the optimal convergence rate of both* EF *and* $\text{EF}^{21}$ *is given by the largest real root of the polynomial*

$$Q(\rho) = \rho^3 - [s(2 + s) + r(s)(sK_1 + K_2)] \rho^2$$
$$+ s^2 [1 + 2s + r(s)(K_1 + sK_2)] \rho - s^4$$

*where $s := \sqrt{\epsilon}$, $r(s) := \frac{(1-s)^2}{1+s}$, and the constants $K_1, K_2$ are defined as*

$$K_1 := \frac{\Delta_2^2 \Sigma_1 + \Delta_1^2 \Sigma_2}{\Sigma_1 \Sigma_2 (\Sigma_1 + \Sigma_2)}, \qquad K_2 := \frac{(\Delta_1 + \Delta_2)^2}{(\Sigma_1 + \Sigma_2)^2},$$

*with $\Sigma_i := L^{(i)} + \mu^{(i)}$ and $\Delta_i := L^{(i)} - \mu^{(i)}$ for $i \in \{1, 2\}$.*

The derivation of this polynomial and a visualization of its roots are provided in Appendix B; see Figure 2. Figure 1 compares the empirical-law rate with the homogeneous worst-case and linear-compressor rates. A further comparison with the distributed $\text{EF}^{21}$ rate of Richtárik et al. (2021) is provided in Appendix B; see Figure 3.

Numerical evidence further suggests a simple tuning rule for the two-parameter EControl method (Gao et al., 2023), where $\gamma$ denotes the model step size and $\eta$ denotes the coefficient on the error term.

**Empirical Law 4.4** (EControl tuning). [PEP]
*Consider the case of heterogeneous regularity parameters, i.e., $f^{(i)} \in \mathcal{F}_{\mu^{(i)}, L^{(i)}}$. The empirically optimal tuning for* EControl *satisfies*

$$\eta^\star_{\text{EControl}} = 0,$$
$$\gamma^\star_{\text{EControl}} = \left( \frac{2n}{\sum_{i=1}^n (L^{(i)} + \mu^{(i)})} \right) \cdot \left( \frac{1 - \sqrt{\epsilon}}{1 + \sqrt{\epsilon}} \right),$$

*which coincides with the optimal step size in Empirical Law 4.1.*

When $\eta = 0$ in EControl, the method is equivalent to $\text{EF}^{21}$ under a reordering of the corresponding parameter $d_k^{(i)}$, and by instead initializing $d_0^{(i)} = 0$. Consequently, the algorithm inherits the same worst-case convergence rates as $\text{EF}^{21}$.

## 5. Conclusion

This paper provides tight worst-case analyses for distributed error-feedback algorithms in the multi-agent setting. Under homogeneous regularity parameters (the same smoothness and strong-convexity constants across agents), the optimal step sizes and contraction factors for EF and $\text{EF}^{21}$ are independent of the number of workers $n$ and coincide with the corresponding single-agent guarantees.

For $\text{EF}^{21}$ with contractive compression, the analysis allows statistical heterogeneity and yields an optimal Lyapunov function together with a worst-case optimal step size and contraction factor. In contrast, a simple counterexample demonstrates that classic EF fails without additional assumptions when local minimizers differ. Under statistical homogeneity (shared minimizer), we provide a tight analysis for EF and recover the same optimal rate as $\text{EF}^{21}$; for deterministic linear compressors we further obtain sharp

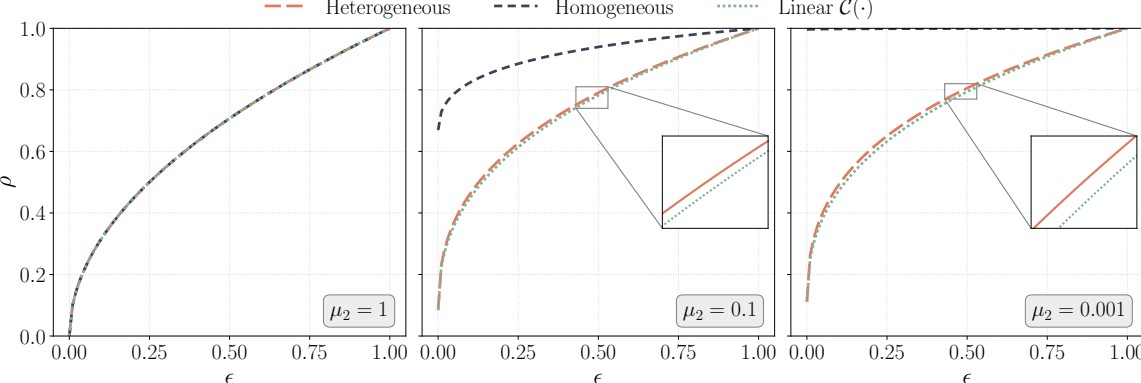

*Figure 1.* Comparison of the rates predicted by Empirical Law 4.3, Theorem 3.1 with worst-case parameters $\max_i L^{(i)}$ and $\min_i \mu^{(i)}$, and Corollary 3.2 with averaged parameters. The inset in the rightmost panel magnifies the gap between the empirical-law rate and the linear-compressor rate. Here $L^{(1)} = L^{(2)} = 1$, $\mu^{(1)} = 1$, and $\mu^{(2)} \in \{1, 0.1, 10^{-3}\}$ from left to right.

guarantees with heterogeneous regularity parameters via an averaged-parameter reduction.

The Lyapunov functions proposed in this analysis could provide insights into the behavior of these methods under different sets of assumptions and, given the close relationship between EF and $\text{EF}^{21}$, could inspire the construction of Lyapunov functions for other error-compensated algorithms. A natural next step is to complement these method-specific tight guarantees with method-agnostic lower bounds, for example lower bounds that depend explicitly on the communication budget or compression level and apply across broader classes of compressed distributed algorithms. These results demonstrate that, when agents share the same smoothness and strong-convexity parameters, the single-agent setting captures the core worst-case behavior and optimal parameter settings observed in larger systems, thereby unifying existing theory.

Finally, a number of empirical laws have been formulated about the behavior of $\text{EF}^{21}$ with general heterogeneous regularity parameters, with strong numerical evidence supporting their validity. Establishing these laws rigorously is left for future work; caution is warranted, as the dependence of the optimal rate on heterogeneous parameters already appears technically intricate, even for $n = 2$.

## Acknowledgments

D. Berg Thomsen and A. Taylor are supported by the European Union (ERC grant CASPER 101162889). The work of A. Dieuleveut is partly supported by ANR-19-CHIA-0002-01/chaire SCAI, and Hi!Paris FLAG project, PEPR Redeem. The French government also partly funded this work under the management of Agence Nationale de la Recherche as part of the "France 2030" program, references ANR-23-IACL-0008 "PR[AI]RIE-PSAI", ANR-23-PEIA-005 (REDEEM project) and ANR-23-IACL-0005.

## Impact Statement

This paper presents work whose goal is to advance the field of Machine Learning. There are many potential societal consequences of this work, none of which the authors feel must be specifically highlighted here.

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

## Organization of the Appendix

This appendix provides additional content and details complementing the paper. In particular, Section A provides the complete missing proofs for the main results of the paper. Section B collects additional empirical-law details, including the Lyapunov discovery pipeline, the derivation and root behavior of Empirical Law 4.3, and a comparison with an existing distributed $EF^{21}$ analysis. Finally, Section C details the verification methodology used to validate the empirical laws.

# A. Proofs

## A.1. Proof of Proposition 2.1

**Proposition 2.1.** *Let there be* $n = 2$ *agents, each having access to first-order oracles querying the one-dimensional quadratic functions defined in* (2)*. Then, for each fixed step size* $\eta > 0$ *in the following cases, there exist admissible compressor responses under Assumption 1.1 for which the full* EF *state* $(x_k, e_k^{(1)}, e_k^{(2)})$ *is 2-periodic:*

1. $\eta < \frac{2}{\mu}$, *with* $x_0 = \eta \frac{\sqrt{\epsilon}}{2 - \eta \mu}$.

2. $\eta > \frac{2}{\mu}$, *with* $x_0 = -\eta \frac{\sqrt{\epsilon}}{2 - \eta \mu}$.

3. $\eta = \frac{2}{\mu}$, *with any* $x_0$.

*Proof.* Begin by dealing with the first case, where $\eta < \frac{2}{\mu}$. Use the same initialization as in (3). Set the compression oracle responses on the EF inputs $e_0^{(i)} + \eta (f^{(i)})'(x_0)$ to be the scaled analogues of the responses in (4). Since the initial error terms are all zero, the first step will be the same as in (4). After the first round of communication, the error terms are given by

$$e_1^{(1)} = e_0^{(1)} + \eta (f^{(1)})'(x_0) - \mathcal{C}(e_0^{(1)} + \eta (f^{(1)})'(x_0); \omega_0^{(1)}) = \sqrt{\epsilon} \eta (f^{(1)})'(x_0),$$
$$e_1^{(2)} = e_0^{(2)} + \eta (f^{(2)})'(x_0) - \mathcal{C}(e_0^{(2)} + \eta (f^{(2)})'(x_0); \omega_0^{(2)}) = -\sqrt{\epsilon} \eta (f^{(2)})'(x_0).$$

Now, set the compression oracles to return the full EF inputs at $x_1$ without compression, i.e., $m_1^{(i)} = e_1^{(i)} + \eta (f^{(i)})'(x_1)$ for $i \in \{1, 2\}$. This will result in the following updates to the error terms:

$$e_2^{(1)} = e_1^{(1)} + \eta (f^{(1)})'(x_1) - \mathcal{C}\left(e_1^{(1)} + \eta (f^{(1)})'(x_1); \omega_1^{(1)}\right) = 0,$$
$$e_2^{(2)} = e_1^{(2)} + \eta (f^{(2)})'(x_1) - \mathcal{C}\left(e_1^{(2)} + \eta (f^{(2)})'(x_1); \omega_1^{(2)}\right) = 0,$$

meaning that the error terms are zero after the second step. This also results in the update

$$\begin{aligned}
x_2 &= x_1 - \frac{1}{2}\left[e_1^{(1)} + \eta (f^{(1)})'(x_1) + e_1^{(2)} + \eta (f^{(2)})'(x_1)\right] \\
&= x_1 - \frac{\eta}{2}\left[\sqrt{\epsilon}(\mu x_0 - 1) + (\mu x_1 - 1) - \sqrt{\epsilon}(\mu x_0 + 1) + (\mu x_1 + 1)\right] \\
&= x_1 - \eta\left(\mu x_1 - \sqrt{\epsilon}\right) = x_0,
\end{aligned}$$

where the last step follows from (3) and some basic algebra. Since the iterate is back at the starting point, with the error terms being zero, the cycle is complete.

The second case where $\eta > \frac{2}{\mu}$ is given by the converse argument, with the starting point $x_0 = -\frac{\eta \sqrt{\epsilon}}{2 - \eta \mu}$ and compression oracles that on the first step give the following responses on the EF inputs (recall $e_0^{(i)} = 0$):

$$\mathcal{C}(\eta (f^{(1)})'(x_0); \omega_0^{(1)}) = \eta(1 + \sqrt{\epsilon})(f^{(1)})'(x_0), \qquad \mathcal{C}(\eta (f^{(2)})'(x_0); \omega_0^{(2)}) = \eta(1 - \sqrt{\epsilon})(f^{(2)})'(x_0).$$

The third case is given by simply considering what would happen if all the compression oracles just gave the true gradient updates during all communication rounds. Then there would be no compression nor error feedback, making the iteration equivalent to distributed gradient descent. The cycle follows from the following computations:

$$\begin{aligned}
x_1 &= x_0 - \frac{\eta}{2}\left[(f^{(1)})'(x_0) + (f^{(2)})'(x_0)\right] \\
&= x_0 - \eta \mu x_0 \\
&= -x_0,
\end{aligned}$$

and,

$$x_2 = x_1 - \frac{\eta}{2}\left[(f^{(1)})'(x_1) + (f^{(2)})'(x_1)\right]$$
$$= x_1 - \eta\mu x_1$$
$$= -x_1.$$

$\square$

## A.2. Proof of Theorem 3.1

**Theorem 3.1.** *Let $\epsilon \in (0,1)$ and assume that the compression operator $\mathcal{C}$ satisfies Assumption 1.1. Let $f^{(i)} \in \mathcal{F}_{\mu,L}$ for each $i \in [n]$. Let the step size be given by*

$$\eta^\star = \left(\frac{2}{L+\mu}\right) \cdot \left(\frac{1-\sqrt{\epsilon}}{1+\sqrt{\epsilon}}\right). \tag{7}$$

*Then, the Lyapunov function*

$$\mathcal{V}\left(\xi^{\mathrm{EF}^{21}}, x; f\right) := \frac{\sqrt{\epsilon}}{n}\left\|\sum_{i=1}^{n} \nabla f^{(i)}(x_k)\right\|^2$$
$$+ \sum_{i=1}^{n}\|\nabla f^{(i)}(x_k) - d_k^{(i)}\|^2 \tag{8}$$

*is optimal (i.e., solves (5)) and satisfies*

$$\mathbb{E}_\omega\left[\mathcal{V}\left(\xi_{k+1}^{\mathrm{EF}^{21}}, x_{k+1}; f\right)\right] \le \rho_\star \cdot \mathbb{E}_\omega\left[\mathcal{V}\left(\xi_k^{\mathrm{EF}^{21}}, x_k; f\right)\right]$$

*where the rate is given by*

$$\rho_\star := \sqrt{\epsilon} + \left(\frac{1-\sqrt{\epsilon}}{2}\right)\left(\frac{\kappa-1}{\kappa+1}\right)^2 \Psi(\kappa, \epsilon) \tag{9}$$

*and*

$$\Psi(\kappa, \epsilon) := 1 - \sqrt{\epsilon} + \sqrt{(1+\sqrt{\epsilon})^2 + \sqrt{\epsilon}\,16\frac{\kappa}{(\kappa-1)^2}}.$$

*Finally, the step size in (7) is worst-case optimal for $\mathrm{EF}^{21}$: within the candidate Lyapunov class based on $\xi^{\mathrm{EF}^{21}}$, it achieves the minimal worst-case one-step contraction factor, $\rho_\star$ in (9). This bound is also multi-step tight, meaning that after $k$ iterations the worst-case contraction equals $\rho_\star^k$.*

*Proof.* The proof of the announced convergence rate follows. Denote $g_k^{(i)} := \nabla f^{(i)}(x_k)$, so that Algorithm 2 writes as

$$x_{k+1} = x_k - \eta \cdot \frac{1}{n}\sum_{i=1}^{n} d_k^{(i)}, \qquad d_{k+1}^{(i)} = d_k^{(i)} + \mathcal{C}(g_{k+1}^{(i)} - d_k^{(i)}; \omega_k^{(i)}).$$

Consider the following inequalities, and associate with each of them the assigned multiplier:

$$I_{\mathcal{F}_{\mu,L}}^{(i,1)} := f^{(i)}(x_k) - f^{(i)}(x_{k+1}) + \frac{\|g_{k+1}^{(i)} - g_k^{(i)}\|^2}{2L} + \langle g_k^{(i)}, x_{k+1} - x_k\rangle \qquad : \lambda_{\mathrm{EF}^{21}}$$
$$+ \frac{\mu}{2(1-\mu/L)}\left\|x_k - x_{k+1} - \frac{1}{L}\left(g_k^{(i)} - g_{k+1}^{(i)}\right)\right\|^2 \le 0,$$

$$I_{\mathcal{F}_{\mu,L}}^{(i,2)} := f^{(i)}(x_{k+1}) - f^{(i)}(x_k) + \frac{\|g_k^{(i)} - g_{k+1}^{(i)}\|^2}{2L} + \langle g_{k+1}^{(i)}, x_k - x_{k+1}\rangle \qquad : \lambda_{\mathrm{EF}^{21}}$$
$$+ \frac{\mu}{2(1-\mu/L)}\left\|x_{k+1} - x_k - \frac{1}{L}\left(g_{k+1}^{(i)} - g_k^{(i)}\right)\right\|^2 \le 0,$$

$$I_{\mathcal{C}}^{(i)} := \mathbb{E}_{\omega_k^{(i)}}\left[\|g_{k+1}^{(i)} - d_k^{(i)} - \mathcal{C}(g_{k+1}^{(i)} - d_k^{(i)}; \omega_k^{(i)})\|^2\right] - \epsilon\,\mathbb{E}_{\omega_k^{(i)}}\left[\|g_{k+1}^{(i)} - d_k^{(i)}\|^2\right] \le 0, \qquad : \nu_{\mathrm{EF}^{21}},$$

where $i \in [n]$, $\nu_{\mathrm{EF}^{21}} := 1$, and where $\lambda_{\mathrm{EF}^{21}}$ is defined as

$$\lambda_{\mathrm{EF}^{21}} := \frac{\sqrt{\epsilon}}{\eta^{\star}(L+\mu)}\left[(1-\sqrt{\epsilon})(L-\mu)+(1+\sqrt{\epsilon})\sqrt{(L-\mu)^2 + \frac{16L\mu\sqrt{\epsilon}}{(1+\sqrt{\epsilon})^2}}\right].$$

Summing the inequalities with their multipliers, the following algebraic identity holds:

$$\lambda_{\mathrm{EF}^{21}}\sum_{i=1}^{n}(I_{\mathcal{F}_{\mu,L}}^{(i,1)} + I_{\mathcal{F}_{\mu,L}}^{(i,2)}) + \nu_{\mathrm{EF}^{21}}\sum_{i=1}^{n}I_{\mathcal{C}}^{(i)} = \mathbb{E}_{\omega_k}\left[\mathcal{V}(\xi_{k+1})\right] - \rho\mathcal{V}(\xi_k) + S, \tag{11}$$

where the residual $S$ is a sum of squares defined as

$$S := naS_{\mathrm{mean}} + cS_{\mathrm{var}} + (\rho - \epsilon)S_{\mathrm{mix}},$$

with the components

$$S_{\mathrm{mean}} := \left\|\bar{d}_k + \frac{1}{a}\left[(\epsilon+b)\bar{g}_{k+1} - (\rho+b)\bar{g}_k\right]\right\|^2,$$

$$S_{\mathrm{var}} := \sum_{i=1}^{n}\left\|(g_{k+1}^{(i)} - \bar{g}_{k+1}) - (g_k^{(i)} - \bar{g}_k)\right\|^2,$$

$$S_{\mathrm{mix}} := \sum_{i=1}^{n}\left\|(d_k^{(i)} - \bar{d}_k) + \frac{\epsilon}{\rho-\epsilon}(g_{k+1}^{(i)} - \bar{g}_{k+1}) - \frac{\rho}{\rho-\epsilon}(g_k^{(i)} - \bar{g}_k)\right\|^2,$$

where $\bar{g} := \frac{1}{n}\sum_{i=1}^{n}g^{(i)}$ and $\bar{d} := \frac{1}{n}\sum_{i=1}^{n}d^{(i)}$ denote the averages, and the coefficients are given by

$$a := \rho - \epsilon + \lambda_{\mathrm{EF}^{21}}\eta^2\frac{L\mu}{L-\mu}, \quad b := \frac{\eta\lambda_{\mathrm{EF}^{21}}}{2}\cdot\frac{L+\mu}{L-\mu}, \quad c := \frac{\lambda_{\mathrm{EF}^{21}}}{L-\mu} - \frac{\epsilon\rho}{\rho-\epsilon}.$$

Note that the positivity of $a$ is guaranteed for $\eta^{\star}$, as noted in the single-agent proof in (Berg Thomsen et al., 2025). To see that $c > 0$, set $t := \sqrt{(1+\sqrt{\epsilon})^2 + \frac{16L\mu\sqrt{\epsilon}}{(L-\mu)^2}}$ and $u := 1 - \sqrt{\epsilon} + t$ (so $u \geq 2$). Then

$$(L-\mu)(\rho-\epsilon)c = a_2(u-2)^2 + a_1(u-2) + \frac{\sqrt{\epsilon}}{(L+\mu)^2}\left[(1+\sqrt{\epsilon}+\epsilon)(L-\mu)^2 + 4\sqrt{\epsilon}L\mu\right],$$

with $a_2, a_1 > 0$, hence $c > 0$.

Since $\lambda_{\mathrm{EF}^{21}} \geq 0$, the weighted sum of inequalities (LHS of (11)) is nonpositive. The statement now follows by plugging in $\eta = \eta^{\star}$ and $\rho = \rho_{\star}$ and checking that all coefficients in (11) are nonnegative.

To show tightness of the bound, consider the case where all agents have the same objective function $f^{(i)} = f$. Initialize $d_0^{(i)} = d_0 = \mathcal{C}(\nabla f(x_0); \omega_0)$ and assume identical compression realizations. Then $d_k^{(i)} = d_k$ for all $k$, and the algorithm updates match the single-agent $\mathrm{EF}^{21}$. The single-agent tight lower bound thus limits the multi-agent performance. Furthermore, since this scenario is a worst-case instance, the optimal step size for the single-agent setting is also worst-case optimal for the multi-agent setting. In particular, since the dynamics of the identical-functions case are invariant to the number of agents $n$, the worst-case optimal step size must also be independent of $n$.

The optimality and multi-step tightness of the Lyapunov function follow from the theory of linear dynamical systems. For the methods and function classes considered, the worst-case contraction is governed by the spectral radius of the iteration matrix on quadratic instances. The class of candidate Lyapunov functions matches this spectral behavior, ensuring that the single-step analysis is optimal and remains tight over multiple iterations (i.e., $\mathcal{V}_k \leq \rho^k\mathcal{V}_0$ is achievable). $\qquad\square$

### A.3. Proof of Corollary 3.2

**Corollary 3.2.** *Let $\epsilon \in [0, 1)$ and assume that the compression operator $\mathcal{C}$ satisfies Assumption 1.1 and is deterministic, additive, and positively homogeneous (so $\mathcal{C}(x+y) = \mathcal{C}(x) + \mathcal{C}(y)$ and $\mathcal{C}(\alpha x) = \alpha\mathcal{C}(x)$ for all $\alpha \geq 0$). Let $f^{(i)} \in \mathcal{F}_{\mu^{(i)}, L^{(i)}}$*

*for each $i \in [n]$, and define*

$$\bar{L} := \frac{1}{n} \sum_{i=1}^{n} L^{(i)}, \qquad \bar{\mu} := \frac{1}{n} \sum_{i=1}^{n} \mu^{(i)}.$$

$$\kappa_\Sigma := \frac{\bar{L}}{\bar{\mu}} = \frac{\sum_{i=1}^{n} L^{(i)}}{\sum_{i=1}^{n} \mu^{(i)}}.$$

*Let the step size be given by*

$$\eta^\star = \left( \frac{2}{\bar{L} + \bar{\mu}} \right) \cdot \left( \frac{1 - \sqrt{\epsilon}}{1 + \sqrt{\epsilon}} \right).$$

*Then the Lyapunov function*

$$\mathcal{V}_{\mathrm{lin}}\left( \xi^{\mathrm{EF}^{21}}, x; f \right) := \sqrt{\epsilon} \|\bar{g}_k\|^2 + \|\bar{g}_k - \bar{d}_k\|^2$$

*satisfies the deterministic contraction*

$$\mathcal{V}_{\mathrm{lin}}\left( \xi_{k+1}^{\mathrm{EF}^{21}}, x_{k+1}; f \right) \le \rho_\star \cdot \mathcal{V}_{\mathrm{lin}}\left( \xi_k^{\mathrm{EF}^{21}}, x_k; f \right),$$

*where $\bar{g}_k := \frac{1}{n} \sum_{i=1}^{n} \nabla f^{(i)}(x_k)$, $\bar{d}_k := \frac{1}{n} \sum_{i=1}^{n} d_k^{(i)}$, and the rate $\rho_\star$ is given by* (9) *with $\kappa = \kappa_\Sigma$.*

*Proof.* Recall $f(x) := \frac{1}{n} \sum_{i=1}^{n} f^{(i)}(x)$, and denote $\bar{g}_k := \nabla f(x_k) = \frac{1}{n} \sum_{i=1}^{n} \nabla f^{(i)}(x_k)$. Since each $f^{(i)}$ is $L^{(i)}$-smooth and $\mu^{(i)}$-strongly convex, $f$ is $\bar{L}$-smooth and $\bar{\mu}$-strongly convex with $\bar{L} = \frac{1}{n} \sum_i L^{(i)}$ and $\bar{\mu} = \frac{1}{n} \sum_i \mu^{(i)}$. Consequently, the standard interpolation (cocoercivity) inequalities for $\mathcal{F}_{\bar{\mu}, \bar{L}}$ apply directly to $\bar{g}_k$ and $\bar{g}_{k+1}$.

The averaged dynamics satisfy

$$x_{k+1} = x_k - \eta \, \bar{d}_k,$$

and, due to additivity and positive homogeneity of $\mathcal{C}$,

$$\bar{d}_{k+1} = \frac{1}{n} \sum_{i=1}^{n} d_{k+1}^{(i)} = \frac{1}{n} \sum_{i=1}^{n} \left( d_k^{(i)} + \mathcal{C}\left( g_{k+1}^{(i)} - d_k^{(i)} \right) \right)$$

$$= \bar{d}_k + \mathcal{C}\left( \frac{1}{n} \sum_{i=1}^{n} \left( g_{k+1}^{(i)} - d_k^{(i)} \right) \right) = \bar{d}_k + \mathcal{C}(\bar{g}_{k+1} - \bar{d}_k).$$

Therefore, the barred variables evolve as a single-agent $\mathrm{EF}^{21}$ instance on $f$ with parameters $(\bar{\mu}, \bar{L})$. The conclusion follows by applying Theorems 1–2 in Berg Thomsen et al. (2025) to this averaged single-agent instance. The optimal step size and rate are obtained by substituting $\kappa_\Sigma = \bar{L}/\bar{\mu} = \sum_i L^{(i)} / \sum_i \mu^{(i)}$ into (9). Theorem 3 in Richtárik et al. (2021) establishes that EF and $\mathrm{EF}^{21}$ are equivalent under deterministic, additive, positively homogeneous compressors, so the same rate applies to EF. $\qquad\square$

### A.4. Proof of Theorem 3.4

**Theorem 3.4.** *Let $\epsilon \in (0, 1)$ and assume that the compression operator $\mathcal{C}$ satisfies Assumption 1.1. Let $f^{(i)} \in \mathcal{F}_{\mu, L}$ for each $i \in [n]$, and assume Assumption 3.3.*

*Let the step size be given by $\eta^\star$ as defined in* (7). *Then, the Lyapunov function*

$$\mathcal{V}\left( \xi^{\mathrm{EF}}, x; f \right) := \frac{1}{n\sqrt{\epsilon}} \left\| \sum_{i=1}^{n} e_k^{(i)} \right\|^2 + \sum_{i=1}^{n} \|x_k - x_\star - e_k^{(i)}\|^2. \tag{10}$$

*is optimal and satisfies*

$$\rho_\star(\mathrm{EF}) = \rho_\star,$$

*where $\rho_\star$ is defined in* (9). *Finally, the step size in* (7) *is worst-case optimal for* EF: *within the candidate Lyapunov class based on $\xi^{\mathrm{EF}}$, it achieves the minimal worst-case one-step contraction factor, $\rho_\star$ in* (9). *This bound is also multi-step tight, meaning that after $k$ iterations the worst-case contraction equals $\rho_\star^k$.*

*Proof.* The proof of the announced convergence rate follows. Algorithm 1 can be rewritten as

$$x_{k+1} = x_k - \frac{1}{n}\sum_{i=1}^n m_k^{(i)}, \qquad e_{k+1}^{(i)} = e_k^{(i)} + \eta\nabla f^{(i)}(x_k) - m_k^{(i)},$$

where $g_k^{(i)} := \nabla f^{(i)}(x_k)$, $u_k^{(i)} := e_k^{(i)} + \eta g_k^{(i)}$, and $m_k^{(i)} = \mathcal{C}(u_k^{(i)};\omega_k^{(i)})$. For $z \in \{e, g, u, m\}$, write $\bar{z}_k := \frac{1}{n}\sum_{i=1}^n z_k^{(i)}$. Consider the following inequalities:

$$I_{\mathcal{F}_{\mu,L}}^{(i,1)} := f^{(i)}(x_k) - f^{(i)}(x^\star) - \langle\nabla f^{(i)}(x_k), x_k - x^\star\rangle + \frac{1}{2L}\|\nabla f^{(i)}(x_k)\|^2$$
$$+ \frac{\mu}{2(1-\mu/L)}\left\|x_k - x^\star - \frac{1}{L}\nabla f^{(i)}(x_k)\right\|^2 \le 0, \qquad : \lambda_{\mathrm{EF}}$$

$$I_{\mathcal{F}_{\mu,L}}^{(i,2)} := f^{(i)}(x^\star) - f^{(i)}(x_k) + \frac{1}{2L}\|\nabla f^{(i)}(x_k)\|^2$$
$$+ \frac{\mu}{2(1-\mu/L)}\left\|x_k - x^\star - \frac{1}{L}\nabla f^{(i)}(x_k)\right\|^2 \le 0, \qquad : \lambda_{\mathrm{EF}}$$

$$I_{\mathcal{C}}^{(i)} := (1-\epsilon)\|u_k^{(i)}\|^2 - 2\left\langle u_k^{(i)}, \mathbb{E}_{\omega_k^{(i)}}\left[\mathcal{C}(u_k^{(i)};\omega_k^{(i)})\right]\right\rangle$$
$$+ \mathbb{E}_{\omega_k^{(i)}}\left[\|\mathcal{C}(u_k^{(i)};\omega_k^{(i)})\|^2\right] \le 0, \qquad : \nu_{\mathrm{EF}},$$

where $i \in [n]$, $\nu_{\mathrm{EF}} := 1/\sqrt{\epsilon}$, and $\lambda_{\mathrm{EF}}$ is defined as

$$\lambda_{\mathrm{EF}} := \frac{\eta^\star}{L+\mu}\left[(1-\sqrt{\epsilon})(L-\mu) + (1+\sqrt{\epsilon})\sqrt{(L-\mu)^2 + \frac{16L\mu\sqrt{\epsilon}}{(1+\sqrt{\epsilon})^2}}\right].$$

Set $\rho := \rho_\star$ and $a := (\rho - \sqrt{\epsilon})(1+\sqrt{\epsilon})/\sqrt{\epsilon}$. Summing these inequalities with their multipliers yields the algebraic identity:

$$\lambda_{\mathrm{EF}}\sum_{i=1}^n(I_{\mathcal{F}_{\mu,L}}^{(i,1)} + I_{\mathcal{F}_{\mu,L}}^{(i,2)}) + \nu_{\mathrm{EF}}\sum_{i=1}^n I_{\mathcal{C}}^{(i)} = \mathbb{E}_{\omega_k}\left[\mathcal{V}(\xi_{k+1})\right] - \rho\mathcal{V}(\xi_k) + S,$$

where the residual $S$ is given by

$$S := naS_{\mathrm{mean}} + cS_{\mathrm{var}} + (\rho - \sqrt{\epsilon})S_{\mathrm{mix}} + (\nu_{\mathrm{EF}} - 1)S_{\mathrm{mix\text{-}comp}},$$

with the components

$$S_{\mathrm{mean}} := \left\|\bar{e}_k - \frac{\rho-1}{a}(x_k - x^\star) + \frac{2(\sqrt{\epsilon}-1)}{a(L+\mu)}\bar{g}_k\right\|^2,$$

$$S_{\mathrm{var}} := \sum_{i=1}^n\left\|g_k^{(i)} - \bar{g}_k\right\|^2,$$

$$S_{\mathrm{mix}} := \sum_{i=1}^n\left\|(e_k^{(i)} - \bar{e}_k) - \frac{\sqrt{\epsilon}\,\eta}{\rho - \sqrt{\epsilon}}(g_k^{(i)} - \bar{g}_k)\right\|^2,$$

$$S_{\mathrm{mix\text{-}comp}} := \sum_{i=1}^n\mathbb{E}_{\omega_k}\left[\left\|(u_k^{(i)} - \bar{u}_k) - (m_k^{(i)} - \bar{m}_k)\right\|^2\right].$$

As in the single-agent proof, the positivity of $\rho - \sqrt{\epsilon}$ (hence $a$) for $\eta^\star$ follows directly from the single-agent analysis. Let $s := \sqrt{\epsilon}$ and $t := \sqrt{(L-\mu)^2 + \frac{16L\mu s}{(1+s)^2}}$. This also gives $\nu_{\mathrm{EF}} - 1 = \frac{1-s}{s} > 0$, and

$$c = \frac{2(1-s)\big((1-s)(L-\mu) + (1+s)t\big)}{(1+s)^2(L-\mu)(L+\mu)^2} > 0.$$

To show tightness of the bound, consider the case where all agents have the same objective function $f^{(i)} = f$. With initialization $e_0^{(i)} = e_0 = 0$ and identical compression realizations, $e_k^{(i)} = e_k$ for all $k$, and the algorithm updates match the single-agent EF. The single-agent tight lower bound thus limits the multi-agent performance. Furthermore, since this scenario is a worst-case instance, the optimal step size for the single-agent setting is also worst-case optimal for the multi-agent setting. In particular, since the dynamics of the identical-functions case are invariant to the number of agents $n$, the worst-case optimal step size must also be independent of $n$.

The optimality of the Lyapunov function and tightness over multiple iterations follow from the same argument as in the proof of Theorem 3.1. □

# B. Additional Details for the Empirical Laws

### B.1. Derivation of the $n = 2$ rate law

The cubic polynomial in Empirical Law 4.3 was identified through an iterative workflow combining numerical PEP experiments with symbolic algebra (using Mathematica). We started from the empirical optimal step size and Lyapunov function in Empirical Laws 4.1 and 4.2, and then formed the dual problem of the corresponding performance estimation problem using that Lyapunov function as the error metric.

Large parameter sweeps showed that the optimal dual variables (Lagrange multipliers) differ substantially from the single-agent case. The associated rate is also slightly different. This ruled out a direct transfer of the single-agent proof structure and required searching for new closed-form relations. We tried several structural constraints, reparameterizations, and symbolic-regression heuristics to infer explicit formulas for the multipliers, without success.

Subsequent experiments indicated that the dual linear matrix inequality (LMI) is rank-deficient in the relevant regimes. Exploiting this structure revealed relations between the multipliers and the convergence rate, which eventually led to the cubic characterization in Empirical Law 4.3 for the $n = 2$ case.

Because the LMI is highly intricate, this discovery required multiple rounds of reparameterization and simplification rather than a single linear derivation.

### B.2. Lyapunov-function discovery pipeline

The Lyapunov structures used in these results were found through a numerical certificate-search workflow. Starting with the two-worker case, we solved the SDP-based Lyapunov search using the log-det heuristic (Fazel et al., 2003) and manually introduced sparsification constraints to identify a minimal structure. Since the remaining coefficients were not unique, we fixed a normalization and probed each coefficient's feasible range by alternately maximizing and minimizing it while leaving the other coefficients free. These ranges revealed a stable limiting pattern as $\epsilon \to 1$, matching the simple coefficients used in the final Lyapunov functions. The resulting candidates were then checked numerically across additional parameter sweeps and larger worker counts.

### B.3. Roots of the $n = 2$ rate polynomial

Closed-form expressions for the roots exist in principle, but they are too cumbersome to be informative in the present context and would significantly increase the length of the paper. Figure 2 contains plots of the roots of the cubic polynomial defined in Empirical Law 4.3. The bifurcation plots highlight that the roots vary nontrivially with $\epsilon$ and heterogeneity, so there is no single root expression that can be uniformly exploited to simplify the cubic (e.g., by direct factorization). In particular, no branch yields a compact expression that remains interpretable across all heterogeneity settings.

### B.4. Comparison with an existing $\mathrm{EF}^{21}$ analysis

Figure 3 compares the relative iteration count predicted by Empirical Law 4.3 against the distributed $\mathrm{EF}^{21}$ rate from Richtárik et al. (2021). The $\mathrm{EF}^{21}$ baseline uses the largest step size allowed by that analysis. The vertical axis is $\log(\rho_{\mathrm{EF}^{21}})/\log(\rho_\star)$, where $\rho_\star$ is the rate in Empirical Law 4.3; a value of $1/2$, for example, means that Empirical Law 4.3 reaches a fixed target accuracy in half as many iterations as the corresponding $\mathrm{EF}^{21}$ bound. The corresponding script in the public repository can be reused with other published rates or standard tuning rules.

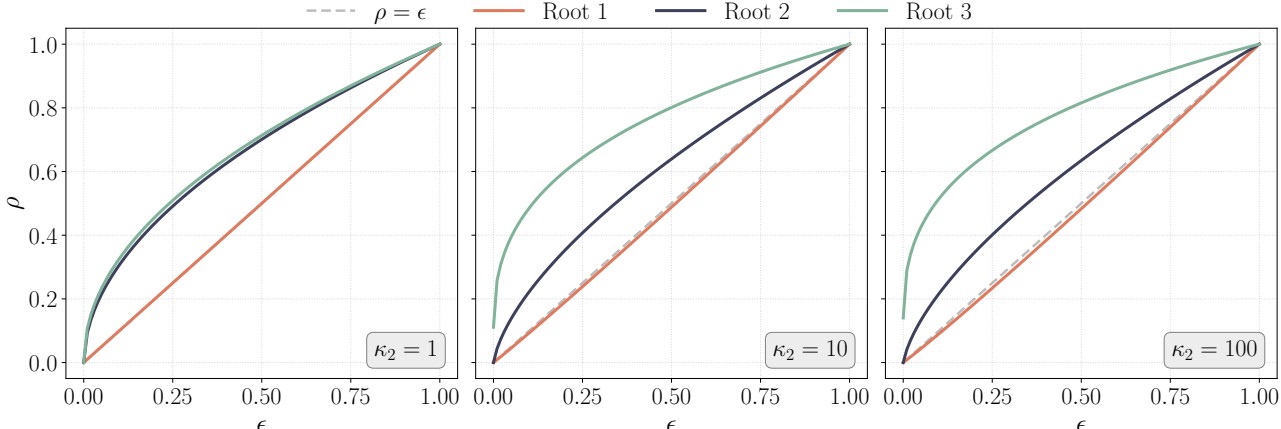

*Figure 2.* Bifurcation plot of the three roots of the cubic polynomial in Empirical Law 4.3 as $\epsilon$ varies. Parameters are $L^{(1)} = L^{(2)} = 1$, $\mu^{(1)} = 0.9$, and $\mu^{(2)} \in \{1, 0.1, 0.01\}$ from left to right. The dashed curve is $\rho = \epsilon$, shown as a reference to indicate that no root branch collapses to this expression uniformly across heterogeneity settings.

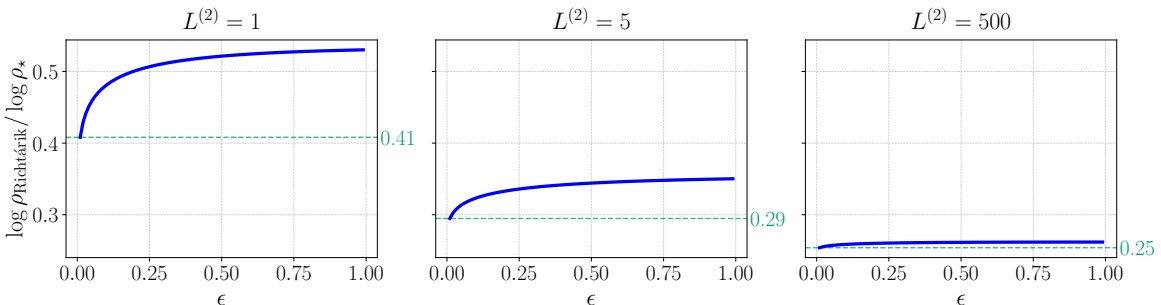

*Figure 3.* Iteration-complexity comparison between the rate $\rho_\star$ from Empirical Law 4.3 and the distributed $\mathrm{EF}^{21}$ rate from Richtárik et al. (2021). The vertical axis is $\log(\rho_{\mathrm{EF}^{21}})/\log(\rho_\star)$, so smaller values indicate fewer iterations for the empirical-law rate to reach a fixed target accuracy. Here $n = 2$, $\mu^{(1)} = \mu^{(2)} = 0.5$, $L^{(1)} = 1$, and $L^{(2)} \in \{1, 5, 500\}$ from left to right.

## C. Verification Details

Validation of the empirical laws was performed using the Performance Estimation Problem (PEP) framework (Drori, 2014; Taylor et al., 2017b). The verifications follow the SDP-based Lyapunov identification techniques of Taylor et al. (2018); Upadhyaya et al. (2025), with the empirical-law certificates implemented directly as CVXPY Lyapunov LMIs in the public notebooks. Some feasibility checks were numerically delicate on highly heterogeneous instances. We therefore accepted only clean solver outputs as certificates: outputs with known numerical warning patterns were treated as uncertified in the corresponding feasibility check. MOSEK (ApS, 2025) was used for bisection and screening, while SDPA (Yamashita et al., 2010) was used as an independent high-precision confirmation step for apparent violations of the empirical laws. Unless stated otherwise, strict MOSEK runs used one thread and interior-point feasibility, gap, and barrier-reduction tolerances $10^{-10}$; SDPA confirmation runs used `epsilonStar`= `epsilonDash` = $10^{-11}$, `mpfPrecision`= 4096, and `maxIteration`= 2000, together with the fixed SDPA tuning parameters reported in the public notebooks.

Across all sweeps, worker permutations were deduplicated because the LMIs are invariant under agent relabeling. For each raw regularity configuration, the smoothness constants were normalized by $L_{\max} = \max_i L^{(i)}$, and the corresponding $\mu^{(i)}$ values were set through $\mu^{(i)} = L^{(i)}/\kappa^{(i)}$ after this normalization. This common rescaling only changes the step-size units and leaves the contraction-factor feasibility unchanged. To test tightness or optimality, we attempted a $1\%$ improvement target, denoted $\rho_{\mathrm{imp}} = 0.99\,\rho$. An apparent improvement was treated as a counterexample only when this improved target was confirmed by the prescribed solver sequence without numerical warnings. No such cleanly confirmed counterexamples were found in any of the experiments below.

## C.1. Verification of Optimal Step Size (Empirical Law 4.1)

The optimal-step experiment was run separately for EF and $EF^{21}$. The EF checks used the homogeneous-worker normalization corresponding to the shared-minimizer setting of Theorem 3.4. Each sweep used 10 linearly spaced values of $\epsilon \in [0.05, 0.95]$. For $n = 2$, we used $L^{(i)} \in \{1, 100, 1000\}$ and $\kappa^{(i)} \in \{2, 100, 1000\}$, giving 450 configurations per method. For $n = 3$, we used $L^{(i)} \in \{1, 1000\}$ and $\kappa^{(i)} \in \{2, 100, 1000\}$, giving 560 configurations per method. For $n = 4$, we used $L^{(i)} \in \{1, 1000\}$ and $\kappa^{(i)} \in \{2, 1000\}$, giving 350 configurations per method.

For each instance, the theoretical step size $\eta^\star$ from Empirical Law 4.1 was evaluated in the full Lyapunov LMI. For $n = 2$, the reference rate $\rho_\star$ was taken from the cubic formula in Empirical Law 4.3 and then certified at $\eta^\star$. For $n = 3, 4$, $\rho_\star$ was computed by bisection on $[0, 1]$ with tolerance $10^{-7}$.

The candidate step sizes are not drawn from the whole interval $[0, 2n/\sum_i(L^{(i)} + \mu^{(i)})]$. Instead, the search is restricted to the band forced by the exact-compression mode. Indeed, if the compressor returns the uncompressed gradient and the error feedback state is zero, both EF and $EF^{21}$ reduce to gradient descent on the averaged objective $\bar{f} := \frac{1}{n}\sum_i f^{(i)}$, whose smoothness and strong-convexity parameters are $\bar{L} := \frac{1}{n}\sum_i L^{(i)}$ and $\bar{\mu} := \frac{1}{n}\sum_i \mu^{(i)}$. The known rate of gradient descent on $\mathcal{F}_{\bar{\mu},\bar{L}}$ is $\max_{\lambda \in [\bar{\mu},\bar{L}]}(1 - \eta\lambda)^2$, so any candidate achieving rate $\rho_\star$ on the full problem class must satisfy

$$\max_{\lambda \in [\bar{\mu},\bar{L}]}(1 - \eta\lambda)^2 \leq \rho_\star, \qquad \text{hence} \qquad \eta \in \left[\frac{1 - \sqrt{\rho_\star}}{\bar{\mu}}, \frac{1 + \sqrt{\rho_\star}}{\bar{L}}\right].$$

A scan of 40 equally spaced values was then performed in this interval, clipped to $[0, 2n/\sum_i(L^{(i)} + \mu^{(i)})]$.

Each candidate step size was first screened with MOSEK at $\rho_{\text{imp}} = 0.99\,\rho_\star$. Candidates that were not certified feasible at this target were discarded. For the remaining candidates, the candidate rate was recomputed by bisection; only candidates still improving on $\rho_\star$ by at least $1\%$ were sent to SDPA for high-precision confirmation. This two-stage procedure avoids false positives from near-boundary numerical noise while keeping the exhaustive search manageable.

## C.2. Verification of Lyapunov Function Structure (Empirical Law 4.2)

This experiment was run for $n \in \{2, 3, 4\}$, with 10 linearly spaced values of $\epsilon \in [0.05, 0.95]$. For each worker, $(L^{(i)}, \kappa^{(i)})$ was chosen from $\{1, 100, 1000\} \times \{2, 100, 1000\}$. This gives 450, 1650, and 4950 configurations for $n = 2, 3, 4$, respectively.

At the conjectured step size $\eta^\star$, bisection on $[0, 1]$ with tolerance $10^{-7}$ was used to compute the rate $\rho_{\text{simp}}$ certified by the simplified Lyapunov structure in Empirical Law 4.2. The full $EF^{21}$ Lyapunov class was then tested at $\rho_{\text{imp}} = 0.99\,\rho_{\text{simp}}$. The first full-class check used MOSEK with strict feasibility and gap tolerances. Whenever this check suggested an improvement, the simplified structure was directly retested at $\rho_{\text{imp}}$ and at a further $10^{-4}$ relative margin; only remaining candidates were confirmed with a high-precision SDPA solve. No SDPA-confirmed full-class improvement over the simplified structure was found.

## C.3. Verification of $n = 2$ Rate (Empirical Law 4.3)

For the $n = 2$ rate law, the sweep used 50 linearly spaced values of $\epsilon \in [0.05, 0.95]$. For each worker, the unscaled parameters were drawn from $L^{(i)} \in \{1, 10, 100, 1000\}$ and $\kappa^{(i)} \in \{2, 10, 100, 1000, 10000\}$, giving 210 unordered regularity configurations and 10500 epsilon-configuration pairs. The candidate rate $\rho_\star$ was taken as the largest valid real root of the cubic in Empirical Law 4.3, and the step size was set to $\eta^\star$.

The empirical law concerns $EF^{21}$; as an auxiliary check, the same candidate was also tested for EF using the homogeneous-worker normalization. For each method, the pair $(\rho_\star, \eta^\star)$ was certified by the corresponding Lyapunov LMI using MOSEK with strict $10^{-10}$ feasibility and gap tolerances. As a local sharpness check, the stronger target $(0.99\,\rho_\star, \eta^\star)$ was also tested and was not certified without numerical warnings on any instance.

## C.4. Verification of EControl Tuning (Empirical Law 4.4)

For EControl, the experiment considered $n \in \{2, 3, 4\}$, with 10 linearly spaced values of $\epsilon \in [0.05, 0.95]$, $L^{(i)} \in \{1, 100, 1000\}$, and $\kappa^{(i)} \in \{2, 100, 1000\}$. This gives 450, 1650, and 4950 configurations for $n = 2, 3, 4$, respectively. For each instance, the conjectured tuning $(\eta, \gamma) = (0, \gamma^\star_{\text{EControl}})$ was evaluated by bisection on the full EControl Lyapunov LMI with tolerance $10^{-7}$, using MOSEK with strict feasibility and gap tolerances.

To search for counterexamples, we tested whether any grid point in the search region could certify a rate at least $1\%$ smaller than the reference rate. Writing this target as $\rho_{\text{imp}} = 0.99\,\rho_\star$, the notebook scanned a $10 \times 10$ grid. The $\eta$-grid was drawn in the unnormalized scale over

$$\eta \in \left[0, \frac{2n}{\sum_i (L_{\text{raw}}^{(i)} + \mu_{\text{raw}}^{(i)})}\right],$$

where $\mu_{\text{raw}}^{(i)} = L_{\text{raw}}^{(i)}/\kappa^{(i)}$. The $\gamma$-grid used the exact-compression band at the improved target,

$$\gamma \in \left[\frac{1 - \sqrt{\rho_{\text{imp}}}}{\bar{\mu}}, \min\left\{\gamma_{\max}, \frac{1 + \sqrt{\rho_{\text{imp}}}}{\bar{L}}\right\}\right],$$

where $\bar{L}$ and $\bar{\mu}$ are computed after max-$L$ normalization and $\gamma_{\max} = L_{\max}\,2n/\sum_i (L_{\text{raw}}^{(i)} + \mu_{\text{raw}}^{(i)})$. Empty $\gamma$-bands were skipped. Each candidate was first screened on the full EControl LMI with MOSEK at $\rho_{\text{imp}}$; any apparent improvement was then re-evaluated by bisection and confirmed with a high-precision SDPA solve. No SDPA-confirmed improvement was found in any tested instance.

