# OpenReview forum: "A Tight Theory of Error Feedback Algorithms in Distributed Optimization"
_ICML.cc/2026/Conference — ICML 2026 regular_

### Official Review · Reviewer_tte7 · 2026-03-11

**Soundness:** 3
**Presentation:** 3
**Significance:** 3
**Originality:** 3
**Overall Recommendation:** 4
**Confidence:** 4

**Summary:**

This paper presents a tight and unifying analysis of popular EF schemes (EF and EF21) for distributed optimization with compressed communications. By choosing the "optimal" Lyapunov functions, stepsizes, etc., the paper gives better convergence bounds than existing work on EF. The "optimality" is confirmed using several techniques, such as PEP/CAS certificates or numerical results (empirical laws).

**Compliance With Llm Reviewing Policy:**

Affirmed.

**Final Justification:**

All my comments have been addressed and I would like to maintain my positive score.

**Key Questions For Authors:**

- To my understanding here, the "optimality" here is by choosing certain parameters (e.g., parameters in the Lyapunov functions and stepsizes) in an optimal way within existing convergence analysis framework. The "optimality" here is NOT established by proving a lower bound, e.g., minimum error given certain communication budget, and show an algoroithm with matching convergence bound.
- The improvement by the proposed "optimal" algorithm over existing analysis of EF schemes should be more clearly highlighted, both theoretically and through numerical experiments.
- Laurent Lassard has a series of papers on achieving better convergence bounds by designing "optimal" Lyapunov functions (often acheived by solving an SDP), which is very similar to the approach in this paper. Yet no detailed comparison is provided at the technical level.

**Strengths And Weaknesses:**

Strength:
- Communication cost is an important topic extensively studied. Despite a variety of EF schemes, the fundamental performance limits of compressed communication is still not well understood.
- Compresive "optimal" convergence bounds for various EF schemes.

Weakness:
- To my understanding here, the "optimality" here is by choosing certain parameters (e.g., parameters in the Lyapunov functions and stepsizes) in an optimal way within existing convergence analysis framework. The "optimality" here is NOT established by proving a lower bound, e.g., minimum error given certain communication budget, and show an algoroithm with matching convergence bound.
- The improvement by the proposed "optimal" algorithm over existing analysis of EF schemes should be more clearly highlighted, both theoretically and through numerical experiments.
- Laurent Lassard has a series of papers on achieving better convergence bounds by designing "optimal" Lyapunov functions (often acheived by solving an SDP), which is very similar to the approach in this paper. Yet no detailed comparison is provided at the technical level.

---

> ### Author Rebuttal · Authors · 2026-03-31
>
> We thank the reviewer for the careful reading and thoughtful comments.
>
> > W1. On our notion of "optimality"
>
> We agree with this distinction, and this is the sense in which "optimality" is used throughout the paper. Specifically, for EF and EF21, within the candidate Lyapunov/state class defined in Section 2.4, we identify the best step size, Lyapunov function, and corresponding tightest worst-case contraction factor, rather than a method-agnostic communication-complexity lower bound over all communication-efficient methods.
>
> Despite being method-specific, the remarks at the end of the proofs of Theorems 3.1 and 3.4 show a relative strengthening of the usual method-specific notion of optimality:
> 1. If one considers a multi-iteration inequality of the form $\mathcal{V}(\xi_{k+T}) \leq \rho \cdot \mathcal{V}(\xi_k)$ for any $T > 0$, this does not improve on the rate we obtain, raised to the power $T$.
> 2. We formally show, without relying on numerical results, that no alternative candidate Lyapunov function yields a better convergence rate.
>
> This follows by viewing the lower bound in [1] as a linear system and applying standard results from control theory. We agree that this point should be stated more prominently when the claims are first introduced. In the revision, we will clarify it earlier (in the introduction) and better explain how it differs from standard method-specific lower-bound statements. We also consider finding a method-agnostic lower bound to be an interesting direction for future research and will mention it in the conclusion.
>
> > W2. On comparing our analysis.
>
> This is a fair request. We ran experiments comparing the rate we provide in Empirical Law 4.3 with existing rates. We provide one example of such a comparison below, against the corresponding rates provided in [2]:
>
> https://github.com/icml2026errorfeedback/icml2026/blob/main/figures/richtarik_multiagent_log_complexity_L_heterogeneity.pdf
>
> The way to interpret the y-axis is that if $\\rho_{\textrm{Richtárik}}=0.99$, and $\\rho_\\star=0.98$, then the value will be 0.5 because it will take twice the number of iterations for the numerator to achieve the same accuracy as the denominator. As you can see, our rate is always better, often by a strong margin.
>
> We will add this comparison, as well as the code generating those plots, enabling easy comparison to any other rate in the literature from existing analyses.
>
> > W3. Laurent Lessard has a series of papers on achieving better convergence bounds by designing "optimal" Lyapunov functions (often achieved by solving an SDP), which is very similar to the approach in this paper. Yet no detailed comparison is provided at the technical level.
>
> Yes: this line of work is one of the main technical ingredients behind our approach. One of the seminal papers on the topic involving Laurent Lessard is [3], which we cite. The contribution of our paper is to bring this perspective to the distributed optimization setting; prior, the closest related result in the error-feedback literature was the single-agent analysis in [1].
>
> The line of work on optimal Lyapunov functions with PEPs was started in [3] and further developed, e.g., in [4] (also cited). Lessard also has a series of papers on Lyapunov analysis [3,5] ([5] being framed from a control-theoretic perspective and not explicitly mentioning Lyapunov functions) and distributed optimization [6,7] (with a decentralized, communication-graph-based flavor and no compression schemes). Overall, [3] is the most directly related reference to what we use, but we will add a mention to [6,7] as well.
>
> In the revision, we will add an appendix subsection explaining more concretely how the Lyapunov functions were obtained, including the framework used to numerically search for them and how we infer simple closed-form expressions from the resulting numerical solutions.
>
> - [1] D. Berg Thomsen et al. *Tight analyses of first-order methods with error feedback*. NeurIPS, 2025.
> - [2] P. Richtárik et al. *EF21: A New, Simpler, Theoretically Better, and Practically Faster Error Feedback*. NeurIPS, 2021.
> - [3] A. Taylor et al. *Lyapunov Functions for First-Order Methods: Tight Automated Convergence Guarantees*. ICML, 2018.
> - [4] M. Upadhyaya et al. *Automated tight Lyapunov analysis for first-order methods*. Math. Prog., 2025.
> - [5] L. Lessard et al. *Analysis and Design of Optimization Algorithms via Integral Quadratic Constraints*. SIAM Optim., 2016.
> - [6] A. Sundararajan et al. *Analysis and Design of First-Order Distributed Optimization Algorithms over Time-Varying Graphs*. IEEE Control Netw. Syst., 2020.
> - [7] A. Sundararajan et al. *A Canonical Form for First-Order Distributed Optimization Algorithms*. ACC, 2019.
> ---
>
> We thank the reviewer for their comments, and hope these clarifications, together with the additional appendix discussion mentioned above, make the scope and positioning of the paper clearer. If so, we would appreciate it if you considered re-evaluating.

---

> > ### Author Rebuttal · Reviewer_tte7 · 2026-04-01
> >
> > All my comments have been addressed.

---

> > > ### Author Response · Authors · 2026-04-07
> > >
> > > Thank you again for taking the time to read our rebuttal and for acknowledging that we resolved all of your comments.
> > >
> > > If your overall assessment of the paper has improved in light of the clarification and additional evidence we provided, we would be grateful if you would consider updating your score to reflect that revised view. Of course, we understand that the final score is entirely at your discretion.
> > >
> > > Thank you again for your time and consideration.

---

### Official Review · Reviewer_tf6q · 2026-03-12

**Soundness:** 3
**Presentation:** 3
**Significance:** 2
**Originality:** 3
**Overall Recommendation:** 4
**Confidence:** 3

**Summary:**

The authors address the convergence analysis of federated learning algorithms based on error-feedback (EF) compression, mainly the algorithms classic EF and EF21. To derive the convergence proof, the authors propose to find the contractive Lyapunov function (within a certain class of candidates) that offers the smallest contraction rate. For some specific settings (e.g., local losses share the same convexity parameters) such best Lyapunov functions are identified, whereas for general heterogeneous cases some empirical ones are indicated.

**Compliance With Llm Reviewing Policy:**

Affirmed.

**Final Justification:**

I increased my score because the authors clarified the main issues I indicated in my original review.

**Key Questions For Authors:**

Q1. In Theorem 3.1, equation (4), can you detail how the proposed Lyapunov function satisfies point 2 of Definition 2.3?

Q2. Is it possible to infer from Theorem 3.1 the rate of convergence of || x_k - x^\star ||, that is, of the distance from the iterate x_k to the solution x^\star of the problem?

Q3. In the paper, several Lyapunov functions are indicated (either claimed to be optimal or only empirical), but the authors don't explain how to arrive at such Lyapunov functions from (2), except in board terms in lines 230-247, second column, e.g., stating that they relied on "multiple tricks." Can you offer more details of such work in an appendix?

**Limitations:**

Yes.

**Strengths And Weaknesses:**

Strengths:
- The authors lay out a novel approach to study the convergence of EF-based algorithms.

Weaknesses:
- The scope of the work is rather limited, since all local functions are assumed to be L-smooth and strongly convex (whereas EF21 applies to nonconvex settings);
- Only the homogeneous setting of shared convex parameters across local functions is solved by the proposed approach (Theorem 3.1);
- It's unclear in practice what is the gain in performance of, say, EF21, when using classical stepsize tuning and the optimal tuning derived in Theorem 3.1; numerical simulations demonstrating the relevance of such gains are needed.

---

> ### Author Rebuttal · Authors · 2026-03-31
>
> We thank the reviewer for the careful reading, for recognizing the novelty of the Lyapunov-based approach, and for highlighting concerns about scope, heterogeneous guarantees, and practical relevance. We address these points in turn below.
>
> > W1. On nonconvex analysis.
>
> Note that while providing worst-case rates for the class of nonconvex functions enables us to cover more problems, an immediate downside would be that the resulting rates would be significantly slower and not necessarily reflective of actual algorithmic behavior. Overall, this is a classical tradeoff in optimization for ML: the most informative class of functions is not always the largest one. A more detailed answer on that point is given in the answer to gmmQ (W2).
>
> > W2. On heterogeneous regularity conditions (and empirical laws).
>
> While some of our general results focus on homogeneous or statistically heterogeneous setups, we also have several insightful results for heterogeneous regularity parameters:
> - Under deterministic, additive, positively homogeneous compression, we provide a formal result in Cor. 3.2.
> - Section 4 provides results in the more generic framework. Although these results are "empirical laws," they are based on extensive numerical experiments that we believe are highly informative. These are not merely simulations on a few benchmark problems: across tens of thousands of configurations, the closed-form formula we provide *exactly matches* the numerical value of the exact worst-case rate.
> - Finally, we believe that obtaining formal proofs of these empirical laws might be extremely hard: e.g., for $n=2$, the general heterogeneous EF21 rate is stated in the paper as the largest root of an explicit cubic polynomial (Empirical Law 4.3). Note that *writing down* the closed form for that root is already a very nontrivial task. We included a bifurcation plot in Appendix B to show that this polynomial cannot simply be reduced by cancelling a trivial root.
>
> In summary, we thus believe those empirical laws are both informative, and "the best we can hope for".
>
> > W3. Gain in performance vs. classical tunings.
>
> We kindly refer to our answer to W2 in tte7. In short, we will add such plots comparing our rates to other analyses and classical tunings.
>
> > Q1. On point 2 of Definition 2.3
>
> Yes. The Lyapunov in (4) is
>
> $$ \\mathcal{V}(\\xi_k) = \\frac{\\sqrt{\\varepsilon}}{n}\\left\\|\\sum_{i=1}^n \\nabla f^{(i)}(x_k)\\right\\|^2 + \\sum_{i=1}^n \\|\\nabla f^{(i)}(x_k)-d_k^{(i)}\\|^2. $$
>
> If $\\mathcal{V}(\\xi_k)=0$, then both nonnegative terms must be zero. The first gives
>
> $$ \\sum_{i=1}^n \\nabla f^{(i)}(x_k)=0. $$
>
> that is, $\\nabla f(x_k)=0$ for the averaged objective $f=\\frac{1}{n}\\sum_i f^{(i)}$. Since $f$ is $\\mu$-strongly convex, it has a unique stationary point, so necessarily $x_k=x_\\star$. The second term then gives
>
> $$ d_k^{(i)}=\\nabla f^{(i)}(x_\\star), \\qquad \\forall i. $$
>
> Hence $\\xi_k^{\\mathrm{EF21}}=\\xi_\\star^{\\mathrm{EF21}}$ uniquely (the converse is immediate, since both terms vanish at that point), with $\\xi_k^{\\mathrm{EF21}}$ given in 3.1 and $\\xi_\\star^{\\mathrm{EF21}} := ([0, \\dots, 0], [\\nabla f^{(1)}(x_\\star), \\dots, \\nabla f^{(n)}(x_\\star)], [\\nabla f^{(1)}(x_\\star), \\dots, \\nabla f^{(n)}(x_\\star)])^\\top$.
>
>
> > Q2. On $\\|x_k-x^\\star\\|$.
>
> Yes. Theorem 3.1 implies a linear rate for the distance to the minimizer. Indeed,
>
> $$
> \\mathcal{V}(\\xi_k) \\;\\ge\\; \\sqrt{\\varepsilon}\\, n \\|\\nabla f(x_k)\\|^2,
> $$
>
> where again $f=\\frac{1}{n}\\sum_i f^{(i)}$. Since $f$ is $\\mu$-strongly convex,
>
> $$
> \\|\\nabla f(x_k)\\| \\ge \\mu \\|x_k-x_\\star\\|,
> $$
>
> and therefore
>
> $$
> \\mathcal{V}(\\xi_k) \\;\\ge\\; \\sqrt{\\varepsilon}\\, n \\mu^2 \\|x_k-x_\\star\\|^2.
> $$
>
> Combining this with $\\mathcal{V}(\\xi_k) \\le \\rho_\\star^k \\mathcal{V}(\\xi_0)$ yields
>
> $$
> \\|x_k-x_\\star\\|^2 \\le \\frac{\\mathcal{V}(\\xi_0)}{\\sqrt{\\varepsilon}\\, n \\mu^2}\\,\\rho_\\star^k.
> $$
> So the iterate distance contracts linearly with the same rate $\\rho_\\star$ (with a different initial constant). This contraction determines the asymptotic rate of convergence of the method, which is what we care about here.
>
> > Q3. On Lyapunov functions.
>
> Yes, and we agree that this part should be explained much more clearly. In the revision, we will add an appendix subsection describing the discovery pipeline more concretely, including both the framework used to numerically search for Lyapunov functions and how we then infer simple closed-form expressions from those numerical solutions.
>
> ---
>
> We hope these clarifications, together with the revisions we commit to here, make the paper's scope, the hierarchy between the rigorous and empirical heterogeneous results, and the practical message of the tuning laws clearer. If they address your concerns, we would be very grateful if you would consider updating your assessment.

---

> > ### Author Rebuttal · Reviewer_tf6q · 2026-04-03
> >
> > The authors have resolved the issues I brought up. I will increase my score.

---

### Official Review · Reviewer_gmmQ · 2026-03-13

**Soundness:** 3
**Presentation:** 3
**Significance:** 3
**Originality:** 3
**Overall Recommendation:** 5
**Confidence:** 4

**Summary:**

In the context of multi-agent distributed optimization, this paper provides a rigorous theoretical analysis of error feedback mechanisms, especially the classical Error Feedback (EF) and the modern EF21 algorithm. This work provides a clear answer to the worst-case convergence rates of these mechanisms under the standard assumptions of compression and strongly convex, smooth objective functions. Experimental results rigorously prove that the theoretically derived convergence rate is tight and cannot be further improved.

**Compliance With Llm Reviewing Policy:**

Affirmed.

**Final Justification:**

My concerns  have been addressed, I consider to raise my score accordingly.

**Key Questions For Authors:**

1. What guidance can these Empirical Laws (4.1-4.4) provide for choosing step sizes or predicting convergence in practical scenarios?
2. The method relies on solving the SDP to identify the optimal Lyapunov function. As the number of agents grows, the dimensionality of the SDP will most likely grow as well, what is the computational limit of your method? For what maximum value of n can you still solve SDP reliably?
3. The analysis of the paper relies on Assumption 1.1. However, many widely used compressors in practice (e.g., unbiased stochastic quantization like QSGD) do not satisfy this assumption. Can your Lyapunov method and convergence be extended to unbiased compressors?

**Limitations:**

yes

**Strengths And Weaknesses:**

Strength

1. This work finds theoretical limits by solving optimal Lyapunov functions, giving tight worst-case rate that can no longer be improved.
2. It successfully generalizes tight convergence analyses, which was previously limited to single agent, to more realistic multi-agent scenarios.
3. Empirical Law 4.1 is based on the harmonic average of the parameters of each intelligent agent. Empirical Law 4.2 indicates that the bottleneck of the system lies in the node with the worst condition number, and optimization should give priority to these nodes. These conclusions have guiding significance for practical applications.

Weakness

1. In Section 4, this paper presents several empirical laws, which are based on extensive numerical experiments (PEP) rather than rigorous mathematical proofs.
2. All analyses in the paper assume that the objective function is strongly convex and smooth. This is the classical assumption, but it also limits its scope of application.
3. The paper distinguishes between statistical heterogeneity and regular heterogeneity, but in the final theoretical results, strong assumptions have to be introduced in some parts in order to obtain rigorous proofs, which means that the paper does not provide a strictly proven, closed-form convergence rate in the general heterogeneous scenario.

---

> ### Author Rebuttal · Authors · 2026-03-30
>
> We thank the reviewer for the careful reading and positive assessment. Below we clarify the scope of Section 4 and answer the 3 questions.
>
> > W1/W3. On empirical laws and heterogeneous regularity.
>
> Please see tf6q (W2). In short we have results on heterogeneous regularities, some indeed based on empirical laws - but those are far stronger than conjectures and demonstrate perfect match between our formulas and numerics in contexts where proofs seem extremely difficult to obtain.
>
> > W2. On nonconvexity.
>
> While practical optimization problems are nonconvex, simpler convex models can still capture meaningful aspects of optimization behavior. Recent work [1] is particularly relevant here: it shows both that learning-rate schedules for large-model training can behave surprisingly similarly to a convex optimization bound, and that this picture can be exploited for practical tuning.
>
> More broadly, many practical methods, including Adam and its variants, were originally designed and understood through much simpler setups. Clean understanding of those simpler roots is often what makes the design of practical algorithms possible in the first place. Convex models can therefore still provide actionable insight, and in our setting they are precisely the regime where PEP/Lyapunov tools yield tight rates, optimal Lyapunov functions, and optimal tuning.
>
> On the distributed side, the paper also broadens scope substantially: previous *tight* convex analyses were limited to the single-agent case, whereas here we study genuinely distributed and heterogeneous multi-agent dynamics. Even under smooth strongly-convex assumptions, this already reveals new behavior: heterogeneity changes the optimal tuning, EF requires additional assumptions while EF21 does not, and the optimal Lyapunov functions are non-trivial. We therefore view the paper as complementary to broader nonconvex analyses: those broaden assumptions, while our contribution is a sharper worst-case characterization in the regime where exactness is possible.
>
> > Q1. On practical guidance from Empirical Laws 4.1-4.4.
>
> We believe they give concrete worst-case tuning guidance, for example:
>
> 1. **Empirical Law 4.1.** In the heterogeneous case, both EF and EF21 should use the step size $$
> \eta^\star=\frac{2n}{\sum_{i=1}^n(L^{(i)}+\mu^{(i)})}\left(\frac{1-\sqrt{\varepsilon}}{1+\sqrt{\varepsilon}}\right).$$
>
>    Compression shrinks the step size by the same factor as in the homogeneous theory; heterogeneity enters only through $\sum_i(L^{(i)}+\mu^{(i)})$.
>
> 2. **Empirical Law 4.2.** The relevant EF21 Lyapunov weights are
>
> $$w_i=\frac{L^{(i)}+\mu^{(i)}}{\sum_j(L^{(j)}+\mu^{(j)})}.$$
>
> This indicates that the worst-case metric should pay more attention to agents with larger local regularity scale, which suggests that these agents deserve the most attention in tuning and studying worst-case behavior.
>
> 3. **Empirical Law 4.4.** This gives a simple message for EControl in the regime we study: the empirically optimal tuning is $\eta_{\mathrm{EControl}}^\star=0$, so the method reduces to EF21 with the same optimal step size as in Empirical Law 4.1.
>
> > Q2. On the computational limit of the SDP approach.
>
> In practice, these SDPs are most useful in numerically tractable regimes, where they help identify structure for later theory. The practical limit depends at least as much on conditioning as on $n$: even at $n=2$, extreme compression or worker heterogeneity can make the SDP ill-conditioned and solver outputs noisy. We have moved primarily to using a high-precision SDP solver [2] for the revision.
>
> > Q3. On unbiased compressors such as QSGD.
>
> QSGD does in fact satisfy the guarantees $E[q(x)]=x$ and $E[||q(x)-x||^2]\le w||x||^2$.
> This is referred to as unbiased and relatively bounded variance (U-RBV) (see, e.g., [3, Lemma 3.1]). For any U-RBV compressor, the rescaled operator $C(\cdot)/(w+1)$ is $\epsilon$-contractive, with $\epsilon=1-1/(w+1)$. The rescaling can be absorbed into the step size. Our upper bounds thus do apply to QSGD or U-RBV compressors.
>
> Note, however, that this reduction may not give the tightest bound: unfortunately, PEP-modeling of unbiased and independent compressors (among agents) is not straightforward, and we consider it as interesting (but complex) future work. We numerically investigated unbiased (but not necessarily independent) compressors - the results are identical, and we deal with contractive independent compressors in the paper. The combination is more difficult.
>
> - [1] Schaipp, et al. *The Surprising Agreement Between Convex Optimization Theory and Learning-Rate Scheduling for Large Model Training*. ICML, 2025.
>
> - [2] Yamashita, et al.. *Implementation and evaluation of SDPA 6.0*. Optimization Methods and Software, 2003.
>
> - [3] Alistarh et al. *QSGD: Communication-Efficient SGD via Gradient Quantization and Encoding*. NeurIPS, 2017.
>
> ---
> We hope these clarifications address your concerns. If so, we would be grateful if you reconsidered your assessment.

---

> > ### Author Rebuttal · Reviewer_gmmQ · 2026-04-03
> >
> > Thank you for the response, my concerns  have been addressed, I consider to raise my score accordingly.

---

### Official Review · Reviewer_1431 · 2026-03-13

**Soundness:** 3
**Presentation:** 3
**Significance:** 3
**Originality:** 3
**Overall Recommendation:** 5
**Confidence:** 1

**Summary:**

This paper provides a convergence analysis for two error feedback algorithms: Error Feedback and EF^{21} by identifying the optimal step size and Lyapunov function.

**Compliance With Llm Reviewing Policy:**

Affirmed.

**Final Justification:**

My concerns have been adequately addressed.

**Key Questions For Authors:**

I am not sure whether the theoretical results presented in the paper provide meaningful guidance for algorithm design.

**Limitations:**

yes

**Strengths And Weaknesses:**

Strengths:

This paper provided a tight worst-case analysis for error feedback algorithms in distributed settings.

Weaknesses:

No experiments validate the effectiveness of theoretical results.

---

> ### Author Rebuttal · Authors · 2026-03-31
>
> We thank the reviewer for the comments.
>
> > No experiments validate the effectiveness of theoretical results.
>
> The goal of this paper was to provide theory for the best-possible worst-case convergence rates of the most common algorithms in this field. The experiments in the appendix confirm our theoretical results because they are not computed on individual examples of smooth strongly-convex functions, but rather over the entire class of smooth strongly-convex functions using the performance estimation problem (PEP) methodology.
>
> One could also run a toy experiment on a set of, e.g., quadratic functions using real or synthetic datasets, but this would deviate somewhat from the stated goal of the paper, which is primarily a theoretical contribution.
>
> For experiments related to the performance of EF and EF21 on nonconvex logistic regression problems, we refer to [1]. We would also be happy to add a sentence to the introduction of the paper highlighting this reference as a place where one can find such results.
>
> > I am not sure whether the theoretical results presented in the paper provide meaningful guidance for algorithm design.
>
> The worst-case behavior of a method on a specific class of optimization problems is a simplified model of the scenarios observed in practice. One direct reason for such a simplification is that it may simply be difficult to provide theory that applies across many different problems without such simplifying assumptions. Yet, when we have theory that *exactly* describes the behavior of methods in those situations, it becomes possible to make fair comparisons between methods. It can indeed provide strong inspiration for algorithm design when one begins to understand what makes such methods perform well relative to one another in this setting.
>
> In our setting, we showed that the optimal step sizes consistently shrink by the compression-dependent factor $\frac{1-\sqrt{\epsilon}}{1+\sqrt{\epsilon}}$, so the theory here gives an explicit tuning rule as a function of the compression level between the agents and the server. Under heterogeneity, we also show that the right tuning scale is governed by averaged local parameters rather than by treating each worker separately, which could provide useful intuition for researchers looking to design new algorithms.
>
> Finally, we would like to highlight that we were able to determine numerically that, in the regime we study, the optimal tuning for EControl effectively reduces to EF21. This is strong evidence that mixing both types of error feedback does not lead to improved convergence guarantees in our setting, which we believe is a strong signal for researchers investigating what mechanisms could lead to further improvements in performance.
>
> - [1] P. Richtárik, I. Sokolov, I. Fatkhullin. *EF21: A New, Simpler, Theoretically Better, and Practically Faster Error Feedback*. NeurIPS, 2021.
>
> ---
> Thank you for your comments. If we managed to respond to your concerns adequately, we would appreciate it if you would reconsider your assessment of the paper.

---

> > ### Author Rebuttal · Reviewer_1431 · 2026-04-06
> >
> > Thank you for your detailed response. I don't have further questions.

---

### Decision · Program_Chairs · 2026-04-30

**Decision:**

Accept (regular)

**Comment:**

This work provided tight convergence analyses for error feedback and EF21 by new proposed Lyapunov functions. The theortical contribution of this paper is interesting. All questions have been addressed by rebuttal, so that I recommend acceptance. Furthermore, authors should incorporate the reviewers' suggestions into their revision.